# Make Sure You're Unsure: A Framework for Verifying Probabilistic Specifications

**Leonard Berrada**[*][†]    **Sumanth Dathathri** [*][†]    **Krishnamurthy (Dj) Dvijotham**[*][†]

**Robert Stanforth** [†]  **Rudy Bunel** [†]  **Jonathan Uesato** [†]  **Sven Gowal** [†]  **M. Pawan Kumar** [†]

## Abstract

Most real world applications require dealing with stochasticity like sensor noise or predictive uncertainty, where formal specifications of desired behavior are inherently probabilistic. Despite the promise of formal verification in ensuring the reliability of neural networks, progress in the direction of probabilistic specifications has been limited. In this direction, we first introduce a general formulation of probabilistic specifications for neural networks, which captures both probabilistic networks (e.g., Bayesian neural networks, MC-Dropout networks) and uncertain inputs (distributions over inputs arising from sensor noise or other perturbations). We then propose a general technique to verify such specifications by generalizing the notion of Lagrangian duality, replacing standard Lagrangian multipliers with "functional multipliers" that can be arbitrary functions of the activations at a given layer. We show that an optimal choice of functional multipliers leads to exact verification (i.e., sound and complete verification), and for specific forms of multipliers, we develop tractable practical verification algorithms.

We empirically validate our algorithms by applying them to Bayesian Neural Networks (BNNs) and MC Dropout Networks, and certifying properties such as adversarial robustness and robust detection of out-of-distribution (OOD) data. On these tasks we are able to provide significantly stronger guarantees when compared to prior work – for instance, for a VGG-64 MC-Dropout CNN trained on CIFAR-10 in a verification-agnostic manner, we improve the certified AUC (a verified lower bound on the true AUC) for robust OOD detection (on CIFAR-100) from $0\% \to 29\%$. Similarly, for a BNN trained on MNIST, we improve on the $\ell_\infty$ robust accuracy from $60.2\% \to 74.6\%$. Further, on a novel specification – distributionally robust OOD detection – we improve on the certified AUC from $5\% \to 23\%$.

## 1 Introduction

While neural networks (NNs) have shown significant promise in a wide-range of applications (for e.g., [He et al., 2016, Yu and Deng, 2014]), a key-bottleneck towards their wide-spread adoption in safety-critical applications is the lack of formal guarantees regarding safety and performance. In this direction, there has been considerable progress towards developing scalable methods that can provide formal guarantees regarding the conformance of NNs with desired properties [Katz et al., 2017, Dvijotham et al., 2018b, Raghunathan et al., 2018]. However, much of this progress has been in the setting where the specifications and neural networks do not exhibit any probabilistic behaviour, or is mostly specialized for specific probabilistic specifications [Weng et al., 2019, Wicker et al., 2020].

---

[*]Equal contribution. Authors listed in alphabetical order. Correspondance to `lberrada@deepmind.com`, `sdathath@deepmind.com`, `dvij@cs.washington.edu`.

[†]DeepMind, London, United Kingdom.

35th Conference on Neural Information Processing Systems (NeurIPS 2021).

In contrast, we introduce a general framework for verifying specifications of neural networks that are probabilistic. The framework enables us to handle stochastic neural networks such as Bayesian Neural Networks or Monte-Carlo (MC) dropout networks, as well as probabilistic properties, such as distributionally robust out-of-distribution (OOD) detection. Furthermore, the specification can be defined on the output distribution from the network, which allows us to handle operations such as the expectation on functions of the neural network output.

Probabilistic specifications are relevant and natural to many practical problems. For instance, for robotics applications, there is uncertainty arising from noisy measurements from sensors, and uncertainty regarding the actions of uncontrolled agents (e.g. uncertainty regarding the behaviour of pedestrians for a self-driving vehicle). Often these uncertainties are modelled using a probabilistic approach, where a distribution is specified (or possibly learnt) over the feasible set of events [Thrun et al., 2005]. In such cases, we want to provide guarantees regarding the network's conformance to desired properties in the distributional setting (e.g. given a model of the pedestrian's uncertain behaviour, guarantee that the probability of collision for the autonomous vehicle is small). A more general problem includes scenarios where there is uncertainty regarding the parameters of the distribution used to model uncertainty. Here, in this general setting, we seek to verify the property that the network behaviour conforms with the desired specification under uncertainty corresponding to an entire set of distributions.

The key to handling the aforementioned complexity in the specifications being verified through our framework is the generalization of the Lagrangian duality. Specifically, instead of using the standard Lagrange duality where the multipliers are linear, we allow for probabilistic constraints (constraints between distributions) and use functional multipliers to replace the linear Lagrange multipliers. This allows us to exploit the structure of these probabilistic constraints, enabling us to provide stronger guarantees and facilitates the verification of *verification-agnostic networks* (networks that are not designed to be verifiable). In our paper, we focus on verification-agnostic networks as this is desirable for many reasons, as noted in Dathathri et al. [2020]. To summarize, our main contributions are:

- We derive a general framework that extends Lagrangian duality to handle a wide range of probabilistic specifications. Our main theoretical result (Theorem 1) shows that our approach (i) is always sound and computes an upper bound on the maximum violation of the specification being verified, and (ii) is expressive enough to theoretically capture tight verification (i.e. obtaining both sound and complete verification).
- We develop novel algorithms for handling specific multipliers and objectives within our framework (Propositions 1, 2). This allows us to apply our framework to novel specifications (such as distributionally robust OOD detection, where input perturbations are drawn from entire sets of distributions) by better capturing the probabilistic structure of the problem.
- We empirically validate our method by verifying neural networks, which are verification-agnostic, on a variety of probabilistic specifications. We demonstrate that even with relatively simple choices for the functional multiplier, our method strongly outperforms prior methods, which sometimes provide vacuous guarantees only. This further points towards the potential for significant improvements to be had by developing tractable optimization techniques for more complex and expressive multipliers within our framework.

## 2 Probabilistic Specifications

### 2.1 Notation

Let us consider a possibly stochastic neural network $\phi : \mathcal{X} \to \mathcal{P}(\mathcal{Y})$, where $\mathcal{X}$ is the set of possible input values to the model, $\mathcal{Y}$ is the set of possible output values, and $\mathcal{P}(\mathcal{Y})$ is the set of distributions over $\mathcal{Y}$. We assume that $\mathcal{Y}$ is a subset of $\mathbb{R}^l$ (unless specified otherwise), where $l$ is the number of labels, and the output of the model are logits corresponding to unnormalized log-confidence scores assigned to the labels $\{1, \ldots, l\}$.

The model is assumed to be a sequence of $K$ layers, each of them possibly stochastic. For $k \in \{1, \ldots, K\}$, $\pi_k(x_k | x_{k-1})$ denotes the probability that the output of layer $k$ takes value $x_k$ when its input value is $x_{k-1}$. We write $x_k \sim \pi_k(x_{k-1})$ to denote that $x_k$ is drawn from the distribution over outputs of layer $k$ given input $x_{k-1}$ to layer $k$. We further assume that each $\pi_k(x)$ has the form $\sigma(\tilde{w}x + \tilde{b})$, where $\sigma$ is a non-linear activation function (e.g., ReLU, sigmoid, MaxOut), and $\tilde{w}$ and $\tilde{b}$

are random variables. The stochasticity for layer $\pi_k$ is assumed to be statistically independent of the stochasticity at other layers. For a BNN, $\tilde{w}$ and $\tilde{b}$ follow a diagonal Gaussian distribution (i.e., a Gaussian distribution with a diagonal covariance matrix), and for a MC-Dropout network they follow a Bernoulli-like distribution.

Given a distribution $p_0$ over the inputs $\mathcal{X}$, we use $\phi(p_0)$ to denote (with a slight abuse) the distribution of the random variable $\phi(X_0)$, where $X_0 \sim p_0$.

## 2.2 Problem Formulation.

We now introduce the general problem formulation for which we develop the verification framework.

**Definition 1** (Probabilistic verification problem). *Given a (possibly stochastic) neural network $\phi : \mathcal{X} \to \mathcal{P}(\mathcal{Y})$, a set of distributions over the input $\mathcal{P}_0$ and a functional $\psi : \mathcal{P}(\mathcal{Y}) \mapsto \mathbb{R}$, the probabilistic verification problem is to check that the following is true:*

$$\forall\, p_0 \in \mathcal{P}_0,\ \psi\left(\phi\left(p_0\right)\right) \leq 0. \tag{1}$$

## 2.3 Examples of Specifications

Below we provide examples of probabilistic specifications which are captured by the above problem formulation, and that we further empirically validate our framework on. In Appendix A, we provide further examples of relevant specifications (e.g., ensuring reliable uncertainty calibration) that can be handled by our problem setup.

**Distributionally Robust OOD Detection.** We consider the problem of verifying that a stochastic neural network assigns low confidence scores to all labels for OOD inputs, even in the presence of bounded noise perturbations to the inputs. Given a noise distribution perturbing an OOD image $x_{\mathrm{ood}}$, we require that the expected softmax is smaller than a specified confidence threshold $p_{\max}$ for each label $i$. Since the precise noise distribution is most often unknown, we wish to consider an entire class $\mathcal{P}_{noise}$ of noise distributions. Denoting by $\delta_x$ the Dirac distribution around $x$, the problem is then to guarantee that for every $p_0$ in $\mathcal{P}_0 = \{\delta_{x_{ood}} + \omega : \omega \in \mathcal{P}_{noise}\}$ and for each possible label $i$, $\psi(\phi(p_0)) \coloneqq \mathbb{E}_{y \sim \phi(p_0)}[\mathtt{softmax}\,(y)_i] - p_{\max} \leq 0$. Robust OOD detection under bounded $\ell_\infty$ perturbations as considered in Bitterwolf et al. [2020] is a special case of this problem where $\mathcal{P}_{noise}$ is restricted to a set of $\delta$ distributions over points with bounded $\ell_\infty$ norm.

**Robust Classification.** We also extend the commonly studied robust classification problem [Madry et al., 2017] under norm-bounded perturbations, to the setting of probabilistic neural networks (e.g. BNNs). Define $\mathcal{P}_0$ to be the set of $\delta$ input distributions centered at points within an $\epsilon$-ball of a nominal point $x_{\mathrm{nom}}$, with label $i \in \{1, \dots, l\}$: $\mathcal{P}_0 = \{\delta_x : \|x - x_{\mathrm{nom}}\| \leq \epsilon\}$. For every $p_0 \in \mathcal{P}_0$, we wish to guarantee that the stochastic NN correctly classifies the input, i.e. for each $j$, $\psi(\phi(p_0)) \coloneqq \mathbb{E}_{y \sim \phi(p_0)}[\mathtt{softmax}\,(y)_i - \mathtt{softmax}\,(y)_j] \leq 0$. Note that it is important to take the expectation of the softmax (and not logits) since this is how inference from BNNs is performed.

# 3 The Functional Lagrangian Framework

We consider the following optimization version:

$$\mathtt{OPT} = \max_{p_0 \in \mathcal{P}_0} \psi\left(\phi\left(p_0\right)\right), \tag{2}$$

Having $\mathtt{OPT} \leq 0$ here is equivalent to satisfying specification (1) . However, solving problem (2) directly to global optimality is intractable in general, because it can possibly be a challenging nonlinear and stochastic optimization problem. However, to only verify that the specification is satisfied, it may suffice to compute an upper bound on $\mathtt{OPT}$. Here, we describe how the functional Lagrangian framework allows to derive such bounds by decomposing the overall problem into smaller, easier sub-problems.

## 3.1 General Framework

Let $\mathcal{X}_k$ denote the feasible space of activations at layer $k$, and let $p_k$ denote the distribution of activations at layer $k$ when the inputs follow distribution $p_0$ (so that $p_K = \phi(p_0)$).

**Assumptions.** In order to derive our verification framework, we make the following assumptions:
(A1): $\exists\, l_0 \leq u_0 \in \mathbb{R}^n$ such that for each input distribution $p_0 \in \mathcal{P}_0$, Support $(p_0) \subseteq \mathcal{X}_0 = [l_0, u_0]$.
(A2): Each layer is such that if $x \in \mathcal{X}_k = [l_k, u_k]$, then Support $(\pi_k(x)) \subseteq \mathcal{X}_{k+1} = [l_{k+1}, u_{k+1}]$.

Assumption (A1) is natural since the inputs to neural networks are bounded. Assumption (A2) can be restrictive in some cases: it requires that the layer output is bounded with probability 1, which is not true, for example, if we have a BNN with a Gaussian posterior. However, we can relax this assumption to requiring that the output is bounded with high probability, as in Wicker et al. [2020].

**Functional Lagrangian Dual.** In order to derive the dual, we begin by noting that problem (2) can be equivalently written in the following constrained form:

$$\max_{p_0 \in \mathcal{P}_0, p_1, \ldots, p_K} \boldsymbol{\psi}\,(p_K)\ \text{s.t.}\ \forall\, k \in \{0, \ldots, K-1\}, \forall\, y \in \mathcal{X}_{k+1},\ p_{k+1}\,(y) = \int_{\mathcal{X}_k} \pi_k\,(y|x)\, p_k\,(x)\, dx.$$

For the $k$-th constraint, let us assign a Lagrangian multiplier $\lambda_{k+1}(y)$ to each possible $y \in \mathcal{X}_{k+1}$. Note that $\lambda\,(y)$ is chosen independently for each $y$, hence $\lambda$ is a *functional multiplier*. We then integrate over $y$, which yields the following Lagrangian penalty to be added to the dual objective:

$$-\int_{\mathcal{X}_{k+1}} \lambda_{k+1}\,(y)\, p_{k+1}\,(y)\, dy + \int_{\mathcal{X}_k, \mathcal{X}_{k+1}} \lambda_{k+1}\,(y)\, \pi_k\,(y|x)\, p_k\,(x)\, dx dy. \tag{3}$$

We now make two observations, which are described here at a high level only and are available in more details in appendix B. First, if we sum these penalties over $k$ and group terms by $p_k$, it can be observed that the objective function decomposes additively over the $p_k$ distributions. Second, for $k \in \{1, \ldots, K-1\}$, each $p_k$ can be optimized independently (since the objective is separable), and since the objective is linear in $p_k$, the optimal $p_k$ is a Dirac distribution, which means that the search over the probability distribution $p_k$ can be simplified to a search over feasible values $x_k \in \mathcal{X}_k$. This yields the following dual:

$$\max_{p_K \in \mathcal{P}_K} \left( \boldsymbol{\psi}\,(p_K) - \int_{\mathcal{X}_K} \lambda_K\,(x)\, p_K\,(x)\, dx \right) + \sum_{k=1}^{K-1} \max_{x \in \mathcal{X}_k} \left( \int_{\mathcal{X}_{k+1}} \lambda_{k+1}\,(y)\, \pi_k\,(y|x)\, dy - \lambda_k\,(x) \right)$$

$$+ \max_{p_0 \in \mathcal{P}_0} \int_{\mathcal{X}_0} \left( \int_{\mathcal{X}_1} \lambda_1\,(y)\, \pi_0\,(y|x)\, dy \right) p_0\,(x)\, dx, \tag{4}$$

where we define $\mathcal{P}_K \triangleq \phi(\mathcal{P}_0)$. In the rest of this work, we refer to this dual as $g\,(\lambda)$, and we use the following notation to simplify equation (4):

$$g\,(\lambda) = \max_{p_0 \in \mathcal{P}_0} g_0(p_0, \lambda_1) + \sum_{k=1}^{K-1} \max_{x_k \in \mathcal{X}_k} g_k(x_k, \lambda_k, \lambda_{k+1}) + \max_{p_K \in \mathcal{P}_K} g_K(p_K, \lambda_K). \tag{5}$$

The dual $g\,(\lambda)$ can be seen as a generalization of Lagrangian relaxation [Bertsekas, 2015] with the two key modifications: (i) layer outputs are integrated over possible values, and (ii) Lagrangian penalties are expressed as arbitrary functions $\lambda_k\,(x)$ instead of being restricted to linear functions.

**Main Result.** Here, we relate the functional Lagrangian dual to the specification objective (2).

**Theorem 1.** *For any collection of functions $\lambda = (\lambda_1, \ldots, \lambda_K) \in \mathbb{R}^{\mathcal{X}_1} \times \ldots \times \mathbb{R}^{\mathcal{X}_K}$, we have that $g\,(\lambda) \geq \mathtt{OPT}$. In particular, if a choice of $\lambda$ can be found such that $g\,(\lambda) \leq 0$, then specification (1) is true. Further, when $\psi\,(p_K) = \mathbb{E}_{y \sim p_K}\,[c\,(y)]$, the dual becomes tight: $g\,(\lambda^\star) = \mathtt{OPT}$ if $\lambda^\star$ is set to:*

$$\lambda_K^\star\,(x) = c\,(x)\,; \forall\, k \in \{K-1, \ldots, 1\},\ \lambda_k^\star\,(x) = \mathbb{E}_{y \sim \pi_k(x)}\,\left[\lambda_{k+1}^\star\,(y)\right].$$

*Proof.* We give a brief sketch of the proof - the details are in Appendix B. The problem in constrained form is an infinite dimensional optimization with decision variables $p_0, p_1, \ldots, p_K$ and linear constraints relating $p_k$ and $p_{k+1}$. The Lagrangian dual of this optimization problem has objective $g\,(\lambda)$. By weak duality, we have $g(\lambda) \geq \mathtt{OPT}$. The second part of the theorem is easily observed by plugging in $\lambda^\star$ in $g\,(\lambda)$ and observing that the resulting optimization problem is equivalent to (2). □

**Example.** Let $\mathcal{P}_0$ be the set of probability distributions with mean 0, variance 1, and support $[-1, 1]$, and let $\mathcal{N}_{[a,b]}(\mu, \sigma^2)$ denote the normal distribution with mean $\mu$ and variance $\sigma^2$ with truncated support $[a, b]$. Now consider the following problem, for which we want to compute an upper bound:

$$\texttt{OPT} = \max_{p_0 \in \mathcal{P}_0} \mathbb{E}_{X_1}[\exp(-X_1)] \quad \text{s.t. } X_1|X_0 \sim \mathcal{N}_{[0,1]}(X_0^2, 1) \text{ and } X_0 \sim p_0. \tag{6}$$

This problem has two difficulties that prevent us from applying traditional optimization approaches like Lagrangian duality [Bertsekas, 2015], which has been used in neural network verification Dvijotham et al. [2018b]. The first difficulty is that the constraint linking $X_1$ to $X_0$ is stochastic, and standard approaches can not readily handle that. Second, the optimization variable $p_0$ can take any value in an entire set of probability distributions, while usual methods can only search over sets of real values. Thus standard methods fail to provide the tools to solve such a problem. Since the probability distributions have bounded support, a possible way around this problem is to ignore the stochasticity of the problem, and to optimize over the worst-case realization of the random variable $X_1$ in order to obtain a valid upper bound on $\texttt{OPT}$ as: $\texttt{OPT} \leq \max_{x_1 \in [0,1]} \exp(-x_1) = 1$. However this is an over-pessimistic modeling of the problem and the resulting upper bound is loose. In contrast, Theorem 1 shows that for any function $\lambda : \mathbb{R} \to \mathbb{R}$, $\texttt{OPT}$ can be upper bounded by:

$$\texttt{OPT} \leq \max_{x_1 \in [0,1], p_0 \in \mathcal{P}_0} \exp(-x_1) - \lambda(x_1) + \mathbb{E}_{X_0 \sim p_0}[\mathbb{E}_{X_1|X_0 \sim \mathcal{N}_{[0,1]}(X_0^2, 1)}[\lambda(X_1)]].$$

This inequality holds true in particular for any function $\lambda$ of the form $x \mapsto \theta x$ where $\theta \in \mathbb{R}$, and thus:

$$\begin{aligned}
\texttt{OPT} &\leq \inf_{\theta \in \mathbb{R}} \max_{x_1 \in [0,1], p_0 \in \mathcal{P}_0} \exp(-x_1) - \theta x_1 + \mathbb{E}_{X_0 \sim p_0}[\mathbb{E}_{X_1|X_0 \sim \mathcal{N}_{[0,1]}(X_0^2, 1)}[\theta X_1]], \\
&= \inf_{\theta \in \mathbb{R}} \max_{x_1 \in [0,1], p_0 \in \mathcal{P}_0} \exp(-x_1) - \theta x_1 + \theta \mathbb{E}_{X_0 \sim p_0}[X_0^2], \\
&= \inf_{\theta \in \mathbb{R}} \max_{x_1 \in [0,1]} \exp(-x_1) - \theta x_1 + \theta \approx 0.37.
\end{aligned}$$

Here, our framework lets us tractably compute a bound on $\texttt{OPT}$ that is significantly tighter compared to the naive support-based bound.

## 3.2 Optimization Algorithm

**Parameterization.** The choice of functional multipliers affects the difficulty of evaluating $g(\lambda)$. In fact, since neural network verification is NP-hard [Katz et al., 2017], we know that computing $g(\lambda^\star)$ is intractable in the general case. Therefore in practice, we instantiate the functional Lagrangian framework for specific parameterized classes of Lagrangian functions, which we denote as $\lambda(\theta) = \{\lambda_k(x) = \lambda_k(x; \theta_k)\}_{k=1}^K$. Choosing the right class of functions $\lambda(\theta)$ is a trade-off: for very simple classes (such as linear functions), $g(\lambda(\theta))$ is easy to compute but may be a loose upper bound on (2), while more expressive choices lead to tighter relaxation of (2) at the cost of more difficult evaluation (or bounding) of $g(\lambda(\theta))$.

---

**Algorithm 1** Verification with Functional Lagrangians

---

**Input:** initial dual parameters $\theta^{(0)}$, learning-rate $\eta$, number of iterations $T$.
**for** $t = 0, \ldots, T-1$ **do** {optimization loop}
    **for** $k = 0$ **to** $K$ **do** {potentially in parallel}
        $d_\theta^{(k)} = \nabla_\theta \left[ \max_{x_k} g_k(x_k, \lambda_k, \lambda_{k+1}) \right]$ {potentially approximate maximization}
    **end for**
    $\theta^{(t+1)} = \theta^{(t)} - \eta \sum_{k=0}^K d_\theta^{(k)}$ {or any gradient based optimization}
**end for**
**Return:** Exact value or guaranteed upper bound on $g(\lambda(\theta^{(T)}))$ {final evaluation}

---

**Optimization.** With some abuse of notation, for convenience, we write $g_0(x_0, \lambda_0, \lambda_1) := g_0(p_0, \lambda_1)$ and $g_K(x_K, \lambda_K, \lambda_{K+1}) := g_K(p_K, \lambda_K)$, with $\lambda_0 = \lambda_{K+1} = 0$. Then the problem of obtaining the best bound can be written as: $\min_\theta \sum_{k=0}^K \max_{x_k} g_k(x_k, \lambda_k, \lambda_{k+1})$, where the inner maximizations are understood to be performed over the appropriate domains ($\mathcal{P}_0$ for $x_0$, $\mathcal{X}_k$ for $x_k$,

$l = 1, \ldots, K-1$ and $\mathcal{P}_K$ for $x_K$). The overall procedure is described in Algorithm 1: $\theta$ is minimized by a gradient-based method in the outer loop; in the inner loop, the decomposed maximization problems over the $x_k$ get solved, potentially in parallel. During optimization, the inner problems can be solved approximately as long as they provide sufficient information about the descent direction for $\theta$.

**Guaranteeing the Final Results.** For the final verification certificate to be valid, we do require the final evaluation to provide the exact value of $g(\lambda(\theta^{(T)}))$ or an upper bound. In the following section, we provide an overview of novel bounds that we use in our experiments to certify the final results.

### 3.3 Bounds for Specific Instantiations

The nature of the maximization problems encountered by the optimization algorithm depends on the verification problem as well as the type of chosen Lagrangian multipliers. In some easy cases, like linear multipliers on a ReLU layer, this results in tractable optimization or even closed-form solutions. In other cases however, obtaining a non-trivial upper bound is more challenging. In this section, we detail two such situations for which novel results were required to get tractable bounds: distributionally robust verification and expected softmax-based problems. To the best of our knowledge, these bounds do not appear in the literature and thus constitute a novel contribution.

**Distributionally Robust Verification with Linexp Multipliers.** We consider the setting where we verify a deterministic network with stochastic inputs and constraints on the input distribution $p_0 \in \mathcal{P}_0$. In particular, we consider $\mathcal{P}_0 = \{\mu + \omega : \omega \sim \mathcal{P}_{noise}\}$, where $\mathcal{P}_{noise}$ denotes a class of zero-mean noise distributions that all satisfy the property of having sub-Gaussian tails (this is true for many common noise distributions including Bernoulli, Gaussian, truncated Gaussian):

$$\text{Sub-Gaussian tail:} \quad \forall i, \forall t \in \mathbb{R}, \ \mathbb{E}\left[\exp\left(t\omega_i\right)\right] \leq \exp\left(t^2\sigma^2/2\right).$$

We also assume that each component of the noise $\omega_i$ is i.i.d. The functional Lagrangian dual $g(\lambda)$ only depends on the input distribution $p_0$ via $g_0$, which evaluates to $g_0(p_0, \lambda_1) = \mathbb{E}_{x \sim p_0}[\lambda_1(x)]$. If we choose $\lambda_1$ to be a linear or quadratic function, then $g(\lambda)$ only depends on the first and second moments of $p_0$. This implies that the verification results will be unnecessarily conservative as they don't use the full information about the distribution $p_0$. To consider the full distribution it suffices to add an exponential term which evaluates to the moment generating function of the input distribution. Therefore we choose $\lambda_1(x) = \alpha^T x + \exp\left(\gamma^T x + \kappa\right)$ and $\lambda_2(x) = \beta^T x$. The following result then gives a tractable upper bound on the resulting maximization problems:

**Proposition 1.** *In the setting described above, and with $s$ as the element-wise activation function:*

$$\max_{p_0 \in \mathcal{P}_0} g_0(p_0, \lambda_1) \leq \alpha^T(w\mu + b) + \exp\left(\left\|w^T\gamma\right\|^2 \sigma^2/2 + \gamma^T b + \kappa\right),$$

$$\max_{x \in \mathcal{X}_1} g_1(x, \lambda_1, \lambda_2) \leq \max_{x \in \mathcal{X}_2, z=s(x)} \beta^T(w_2 z + b_2) - \alpha^T x - \exp\left(\gamma^T x + \kappa\right).$$

*The maximization in the second equation can be bounded by solving a convex optimization problem (Appendix C.3).*

**Expected Softmax Problems.** Several of the specifications discussed in Section 2.3 (e.g., distributionally robust OOD detection) require us to bound the expected value of a linear function of the softmax. For specifications whose function can be expressed as an expected value: $\psi(p_K) = \mathbb{E}_{x \sim p_K}[c(x)]$, by linearity of the objective w.r.t. the output distribution $p_K$, the search over the distribution $p_k$ can be simplified to a search over feasible output values $x_K$:

$$\max_{p_K \in \mathcal{P}_K} \psi(p_K) - \int_{\mathcal{X}_K} \lambda_K(x) p_K(x) \, dx = \max_{x \in \mathcal{X}_K} c(x) - \lambda_K(x). \tag{7}$$

Given this observation, the following lets us certify results for linear functions of the $\mathtt{softmax}(x)$:

**Proposition 2.** *For affine $\lambda_K$, and $c(x)$ with the following form $c(x) = \mu^T \mathtt{softmax}(x)$, $\max_{x \in \mathcal{X}_K} c(x) - \lambda_K(x)$ can be computed in time $O(3^d)$, where $\mathcal{X}_K \subseteq \mathbb{R}^d$.*

We provide a proof of this proposition and a concrete algorithm for computing the solution in Appendix C.2. This setting is particularly important to measure verified confidence and thus to perform robust OOD detection. We further note that while the runtime is exponential in $d$, $d$ corresponds to the number of labels in classification tasks which is a constant value and does not grow with the size of the network or the inputs to the network. Further, the computation is embarrassingly parallel and can be done in $O(1)$ time if $3^d$ computations can be run in parallel. For classification problems with 10 classes (like CIFAR-10 and MNIST), exploiting this parallelism, we can solve these problems on the order of milliseconds on a cluster of CPUs.

## 4 Related Work

**Verification of Probabilistic Specifications.** We recall that in our work, $\mathcal{P}_0$ refers to a space of distributions on the inputs $x$ to a network $\phi$, and that we address the following problem: verify that $\forall p \in \mathcal{P}_0, \phi(p) \in \mathcal{P}_{out}$, where $\mathcal{P}_{out}$ represents a constraint on the output distribution. In contrast, prior works by Weng et al. [2019], Fazlyab et al. [2019], and Mirman et al. [2021] study probabilistic specifications that involve robustness to probabilistic perturbations of a single input for deterministic networks. This setting can be recovered as a special case within our formalism by letting the class $\mathcal{P}_0$ contain a single distribution $p$. Conversely, Dvijotham et al. [2018a] study specifications involving stochastic models, but can not handle stochasticity in the input space.

Wicker et al. [2020] define a notion of probabilistic safety for BNNs: $\mathbb{P}_{w \sim p_w} [\forall x \in \mathcal{X}, \phi_w(x) \in \mathcal{C}] \geq p_{\min}$, where $\mathbb{P}$ is the probability of any event, $\phi_w$ denotes the network with parameters $w$, $p_w$ denotes the distribution over network weights (e.g., a Gaussian posterior) and $\mathcal{C}$ is a set of safe outputs, and this allows for computation of the probability that a randomly sampled set of weights exhibits safe behaviour. However, in practice, inference for BNNs is carried out by averaging over predictions under the distribution of network weights. In this less restrictive and more practical setting, it suffices if the constraint is satisfied by the probabilistic prediction that averages over sampled weights: $\forall x \in \mathcal{X}, \mathbb{P}_{w \sim p_w} [\phi_w(x) \in \mathcal{C}] \geq p_{\min}$, where $\phi_w(x)$ denotes the distribution over outputs for $x \in \mathcal{X}$. Further, Wicker et al. [2020] also observe that $\min_{x \in \mathcal{X}} \mathbb{P}_{w \sim p_w} [\phi_w(x) \in \mathcal{C}] \geq \mathbb{P}_{w \sim p_w} [\forall x \in \mathcal{X}, \phi_w(x) \in \mathcal{C}]$, making the second constraint less restrictive. Cardelli et al. [2019] and Michelmore et al. [2020] consider a similar specification, but unlike the approaches used here and by Wicker et al. [2020], these methods can give statistical confidence bounds but not certified guarantees.

Wicker et al. [2021] improve the classification robustness of Bayesian neural networks by training them to be robust based on an empirical estimate of the average upper bound on the specification violation, for a fixed set of sampled weights. In contrast, our approach provides meaningful guarantees for BNNs trained without considerations to make them more easily verifiable, and the guarantees our framework provides hold for inference based on the true expectation, as opposed to a fixed set of sampled weights.

Our work also generalizes Bitterwolf et al. [2020], which studies specifications of the output distribution of NNs when the inputs and network are deterministic. In contrast, our framework's flexibility allows for stochastic networks as well. Furthermore, while Bitterwolf et al. [2020] are concerned with training networks to be verifiable, their verification method fails for networks trained in a verification-agnostic manner. In our experiments, we provide stronger guarantees for networks that are trained in a verification-agnostic manner.

**Lagrangian Duality.** Our framework subsumes existing methods that employ Lagrangian duality for NN verification. In Appendix D.1, we show that the functional Lagrangian dual instantiated with linear multipliers is equivalent to the dual from Dvijotham et al. [2018b]. This is also the dual of the LP relaxation [Ehlers, 2017] and the basis for other efficient NN verification algorithms ([Zhang et al., 2018, Singh et al., 2018], for example), as shown in Liu et al. [2021]. For the case of quadratic multipliers and a particular grouping of layers, we show that our framework is equivalent to the Lagrangian dual of the SDP formulation from Raghunathan et al. [2018] (see Appendix D.2).

We also note that similar mathematical ideas on nonlinear Lagrangians have been explored in the optimization literature [Nedich and Ozdaglar, 2008, Feizollahi et al., 2017] but only as a theoretical construct - this has not lead to practical algorithms that exploit the staged structure of optimization problems arising in NN verification. Further, these approaches do not handle stochasticity.

Table 1: Robust OOD Detection: MNIST vs EMNIST (MLP and LeNet) and CIFAR-10 vs CIFAR-100 (VGG-*). BP: Bound-Propagation (baseline); FL: Functional Lagrangian (ours). The reported times correspond to the median of the 500 samples.

| OOD Task | Model | #neurons | #params | $\epsilon$ | Time (s) | | GAUC (%) | | AAUC (%) |
|---|---|---|---|---|---|---|---|---|---|
| | | | | | BP | FL | BP | FL | |
| (E)MNIST | MLP | 256 | 2k | 0.01 | 40.1 | +38.8 | 55.4 | **65.0** | 86.9 |
| | | | | 0.03 | 40.1 | +37.4 | 38.5 | **53.1** | 88.6 |
| | | | | 0.05 | 38.4 | +36.2 | 18.9 | **36.1** | 88.8 |
| (E)MNIST | LeNet | 0.3M | 0.1M | 0.01 | 53.2 | +52.4 | 0.0 | **29.8** | 71.6 |
| | | | | 0.03 | 52.4 | +51.1 | 0.0 | **14.1** | 57.6 |
| | | | | 0.05 | 55.4 | +54.1 | 0.0 | **3.1** | 44.0 |
| CIFAR | VGG-16 | 3.0M | 83k | 0.001 | 50.8 | +35.0 | 0.0 | **25.6** | 60.9 |
| | VGG-32 | 5.9M | 0.2M | 0.001 | 82.3 | +40.9 | 0.0 | **25.8** | 64.7 |
| | VGG-64 | 11.8M | 0.5M | 0.001 | 371.7 | +48.7 | 0.0 | **29.5** | 67.4 |

## 5 Experiments

Here, we empirically validate the theoretical flexibility of the framework and its applicability to across several specifications and networks. Crucially, we show that our framework permits verification of probabilistic specifications by effectively handling parameter and input stochasticity across tasks. For all experiments, we consider a layer decomposition of the network such that the intermediate inner problems can be solved in closed form with linear multipliers (See Appendix C.4). We use different bound-propagation algorithms to compute activation bounds based on the task, and generally refer to these methods as BP. Our code is available at `https://github.com/deepmind/jax_verify`.

### 5.1 Robust OOD Detection on Stochastic Neural Networks

**Verification Task.** We consider the task of robust OOD detection for stochastic neural networks under bounded $\ell_\infty$ inputs perturbation with radius $\epsilon$. More specifically, we wish to use a threshold on the softmax value (maximized across labels) to classify whether a sample is OOD. By using verified upper bounds on the softmax value achievable under $\epsilon$-perturbations, we can build a detector that classifies OOD images as such even under $\epsilon$ perturbations. We use the Area Under the Curve (AUC) to measure the performance of the detector. Guaranteed AUC (GAUC) is obtained with verified bounds on the softmax, and Adversarial AUC (AAUC) is based on the maximal softmax value found through an adversarial attack. We consider two types of stochastic networks: i) Bayesian neural networks (BNNs) whose posterior distribution is a truncated Gaussian distribution. We re-use the MLP with two hidden layers of 128 units from Wicker et al. [2020] (denoted as MLP) and truncate their Gaussian posterior distribution to three standard deviations, ii) we consider MC-Dropout convolutional neural networks, namely we use LeNet (as in Gal and Ghahramani [2016]) and VGG-style models [Simonyan and Zisserman, 2015]. For MLP and LeNet, we use MNIST as training data, and EMNIST as out-of-distribution data. For the VGG models, we use CIFAR-10 as training data, and CIFAR-100 as out-of-distribution data.

**Method.** We use linear Lagrangian multipliers, which gives closed-form solutions for all intermediate inner maximization problems (Appendix C.4). In addition, we leverage Proposition 2 to solve the final inner problem with the softmax specification objective. Further experimental details, including optimization hyper-parameters, are available in Appendix E.1. We compute activation bounds based on an extension of Bunel et al. [2020] to the bilinear case, referred to as BP in Table 1. The corresponding bounds are obtained with probability 1, and if we were to relax these guarantees to only hold up to some probability lower than 1, we note that the method of [Wicker et al., 2020] would offer tighter bounds.

**Results.** The functional Lagrangian (FL) approach systematically outperforms the Bound-Propagation (BP) baseline. We note that in particular, FL significantly outperforms BP on dropout CNNs, where BP is often unable to obtain any guarantee at all (See Table 1). As the size of the VGG

Table 2: Adversarial Robustness for different BNN architectures trained on MNIST from Wicker et al. [2020]. The accuracy reported for FL and LBP is the % of samples we can certify as robust with probability 1. For each model, we run the experiment for the first 500 test-set samples.

| #layers | $\epsilon$ | #neurons | LBP Acc. (%) | FL Acc. (%) | LBP Time (s) | FL Time (s) |
|---------|-----------|----------|--------------|-------------|--------------|-------------|
| 1 | 0.025 | 128 | 67.0 | **77.2** | 16.7 | +518.3 |
|   |       | 256 | 66.2 | **76.4** | 16.1 | +522.8 |
|   |       | 512 | 60.2 | **74.6** | 16.0 | +522.4 |
| 2 | 0.001 | 256 | 57.0 | **70.0** | 16.8 | +516.5 |
|   |       | 512 | 79.6 | **87.4** | 17.0 | +517.3 |
|   |       | 1024 | 39.4 | **42.4** | 17.1 | +514.1 |

model increases, we can observe that the median runtime of BP increases significantly, while the additional overhead of using FL remains modest.

## 5.2 Adversarial Robustness for Stochastic Neural Networks

**Verification Task.** For this task, we re-use the BNNs trained on MNIST [LeCun and Cortes, 2010] from Wicker et al. [2020] (with the Gaussian posterior truncated to three standard deviations bounded). We use the same setting as Wicker et al. [2020] and study the classification robustness under $\ell_\infty$ perturbations for 1-layered BNNs and 2-layered BNN at radii of $\epsilon = 0.025$ and $\epsilon = 0.001$ respectively. We recall, as mentioned earlier in Section 4, that the specification we study is different from that studied in [Wicker et al., 2020].

**Method** We use the same solving methodology as in Section 5.1. To compute bounds on the activations, we use the bilinear LBP method proposed in Wicker et al. [2020].

**Results.** Across settings (Table 2) our approach is able to significantly improve on the guarantees provided by the LBP baseline, while also noting that our method incurs an increased compute cost.

## 5.3 Distributionally Robust OOD Detection

**Verification Task.** We bound the largest softmax probability across all labels for OOD inputs over noisy perturbations of the input where the noise is drawn from a family of distributions with only two constraints for each $p \in \mathcal{P}_{noise}$: $\omega \in [-\epsilon, \epsilon]$ with prob. $1, \mathbb{E}_{\omega \sim p} \left[ \exp \left( \omega t \right) \right] \leq \exp \left( \sigma^2 t^2 / 2 \right)$, for given constraints $\epsilon, \sigma > 0$. The first constraint corresponds to a restriction on the support of the noise distribution, and the second constraint requires that the noise distribution is sub-Gaussian with parameter $\sigma$. We note that no prior verification method, to the best of our knowledge, addresses this setting. Thus, as a baseline, we use methods that only use the support of the distribution and perform verification with respect to worst-case noise realizations within these bounds.

**Method.** We use a 3-layer CNN trained on MNIST with the approach from Hein et al. [2019] (details in Appendix E.3), and we employ EMNIST as the out-of-distribution data to be detected. We use functional multipliers of the form: $\lambda_k \left( x \right) = \theta_k^T x$ for $k > 1$ and linear-exponential multipliers for the input layer: $\lambda_1 \left( x \right) = \theta_1^T x + \exp \left( \gamma_1^T x + \kappa_1 \right)$ (method denoted by FL-LinExp). As baselines, we consider a functional Lagrangian setting with linear multipliers that only uses information about the expectation of the noise distribution (method denoted by FL-Lin), and a BP baseline that only uses information about the bounds on the support of the noise distribution $(-\epsilon, \epsilon)$ (activation bounds are computed using Bunel et al. [2020]). The inner maximization of $g_k$ for $K - 1 \geq k \geq 2$ can be done in closed form and we use approaches described in Propositions 1 and 2 to respectively solve $\max g_0, \max g_1$ and $\max g_K$. We use $\epsilon = 0.04, \sigma = 0.1$.

**Results.** FL-LinExp achieves the highest guaranteed AUC (See Table 3), showing the value of accounting for the noise distribution, instead of relying only on bounds on the input noise.

Table 3: Guaranteed Area Under Curve (GAUC) values in a distributionally robust setting. The stochastic formulation of the Functional Lagrangian with Linear-Exponential (LinExp) multipliers gets the highest guaranteed AUC.

| Model | #neurons | GAUC (%) | | | Timing (s) | | |
|-------|----------|------|--------|-----------|-------|--------|-----------|
| | | BP | FL-Lin | FL-LinExp | BP | FL-Lin | FL-LinExp |
| CNN | 9972 | 5.4 | 5.6 | **23.2** | 227.7 | +661.8 | +760.7 |

## 6 Conclusion

We have presented a general framework for verifying probabilistic specifications, and shown significant improvements upon existing methods, even for simple choices of the Lagrange multipliers where we can leverage efficient optimization algorithms. We believe that our approach can be significantly extended by finding new choices of multipliers that capture interesting properties of the verification problem while leading to tractable optimization problems. This could lead to discovery of new verification algorithms, and thus constitutes an exciting direction for future work.

**Limitations.** We point out two limitations in the approach suggested by this work. First, the guarantees provided by our approach heavily depend on the bounding method used to obtain the intermediate bounds – this is consistent with prior work on verifying deterministic networks [Dathathri et al., 2020, Wang et al., 2021], where tighter bounds result in much stronger guarantees. Second, as noted in Section 3.1, our approach can only handle probability distributions that have bounded support, and alleviating this assumption would require further work.

**Broader Impact and Risks.** Our work aims to improve the reliability of neural networks in settings where either the model or its inputs exhibit probabilistic behaviour. In this context, we anticipate this work to be largely beneficial and do not envision malicious usage. However, the guarantees of our method crucially depend on having an accurate modeling of the uncertainty. Failing that can result in an overestimation of the reliability of the system, which can have catastrophic consequences in safety-critical scenarios. In this regard, we advocate special care in the design of the specification when applying our approach to real-world use cases.

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
