# Appendix

## A   Probabilistic Specifications: Examples

Below we provide further examples of specifications that can be captured by our framework.

**Uncertainty calibration.**   Another desirable specification towards ensuring reliable uncertainty calibration for NNs is that the expected uncertainty in the predictions increases monotonically with an increase in the variance of the input-noise distribution. Formally, consider a set of zero-mean distributions $\mathcal{P}_{noise}$ with diagonal covariance matrices. For any two such noise distributions $p_{\omega_1}, p_{\omega_2} \in \mathcal{P}_{noise}$ such that $\mathrm{Var}(p_{\omega_1}) \succcurlyeq \mathrm{Var}(p_{\omega_2})$ (where $\succcurlyeq$ is the element-wise inequality and $\mathrm{Var}$ denotes the diagonal of the covariance matrix) and a given image $x$ from the training distribution, we wish to guarantee that the expected entropy of the resulting predictions corresponding to $p_{\omega_1}$ is greater than that of $p_{\omega_2}$, i.e., $\mathbb{E}_{x_1 \sim x + p_{\omega_1}} [H(\texttt{softmax}(\phi(x_1))] \geq \mathbb{E}_{x_2 \sim x + p_{\omega_2}} [H(\texttt{softmax}(\phi(x_2))]$, where $H$ is the entropy functional: $H(p) = -\sum_{i=1}^{|\mathcal{Y}|} p_i \log p_i$. Intuitively, this captures the desired behaviour that as the strength of the noise grows, the expected uncertainty in the network predictions increases. We can capture this specification within the formulation described by equation 1, by letting:

1. $\mathcal{P}_0 = \mathcal{P}_{noise} \times \mathcal{P}_{noise}$,
2. For a given NN $\phi$ and an input $x$, we define another network $\bar{\phi} : \mathcal{X} \times \mathcal{X} \to \mathcal{P}(\mathcal{Y}) \times \mathcal{P}(\mathcal{Y})$, where $\bar{\phi}$ is such that: $\bar{\phi}(a, b) = (\phi(x + a), \phi(x + b))$.

We can then define the verification problem as certifying that the following holds :

$$\psi \left( \bar{\phi}(\bar{p}_0, \bar{\bar{p}}_0) \right) := - \left( \mathbb{E}_{(y_1, y_2) \sim \bar{\phi}(\bar{p}_0, \bar{\bar{p}}_0)} \left[ H \left( \texttt{softmax} \left( y_1 \right) \right) - H \left( \texttt{softmax} \left( y_2 \right) \right) \right] \right) \leq 0,$$

$$\forall \left( \bar{p}_0, \bar{\bar{p}}_0 \right) \in \mathcal{P}_0 \text{ such that } \mathrm{Var}(\bar{p}_0) \succcurlyeq \mathrm{Var}(\bar{\bar{p}}_0).$$

**Robust VAE**   In Dathathri et al. [2020], the authors consider a specification that corresponds to certifying low reconstruction losses for a VAE decoder over a set of inputs in the neighborhood of the latent-variable mean predicted by the encoder for a given image. A natural generalization of this specification is one where low reconstruction error is guaranteed in expectation, since in practice the latent-representations that are fed into the decoder are drawn from a normal distribution whose mean and variance are predicted by the encoder. A more general specification is one where we wish to verify that for a set of norm-bounded points around a given input, the expected reconstruction error from the VAE is small. Formally, for a VAE $\phi$ (note that a VAE directly fits within our framework, where the distribution for the latent-variable can be obtained as the output of a stochastic layer), a given threshold $\tau \in \mathbb{R}_+$ and for a set of inputs $\mathcal{S}$, we wish to certify that the following holds:

$$\forall p_0 \in \mathcal{P}_0, \quad \psi(\phi(p_0)) := \mathbb{E}_{\mu_\phi^{s'} \sim \phi(p_0)} \left[ \| \mathbb{E}_{s \sim p_0} [s] - \mu_\phi^{s'} \|_2^2 \right] - \tau \leq 0; \tag{8}$$

where $\mathcal{P}_0 = \{ \delta_s : s \in \mathcal{S} \}$.

## B   Proof of Functional Lagrangian Theorem

**Proof of Theorem 1**

*Proof.* The optimization problem (2) is equivalent to the following optimization problem:

$$\max_{p_0, p_1, \dots, p_K} \quad \psi(p_K) \tag{9a}$$

$$\text{Subject to } p_{k+1}(y) = \int \pi_k(y|x) \, p_k(x) \, dx \quad \forall y \in \mathcal{X}_{k+1}, \ \forall k \in \{0, \dots, K-1\}$$

$$p_0 \in \mathcal{P}_0 \tag{9b}$$

where $\psi$ is a functional of the output distribution $p_K$. We refer to the space of probability measures on $\mathcal{X}_k$ as $\mathcal{P}_k$ (not that for $k = 0$, this may not be the whole space of probability measures but a

constrained set of measures depending on the specification we would like to verify). The dual of this optimization problem can be written as follows:

$$\max_{p_0 \in \mathcal{P}_0, p_1 \in \mathcal{P}_1, \dots} \boldsymbol{\psi}\left(p_K\right) - \sum_{k=0}^{K-1} \int \lambda_{k+1}\left(z\right) p_{k+1}\left(z\right) dz$$

$$+ \sum_{k=0}^{K-1} \int_{\mathcal{X}_{k+1}, \mathcal{X}_k} \lambda_{k+1}\left(z\right) \pi_k\left(z|x\right) p_k\left(x\right) dx dz,$$

where we assigned Lagrange multipliers $\lambda_{k+1}\left(y\right)$ for every $y \in \mathcal{X}_{k+1}$. The above optimization problem can be decomposed as follows:

$$\boldsymbol{\psi}\left(p_K\right) - \int_{\mathcal{X}_K} \lambda_K\left(x\right) p_K\left(x\right) dx$$

$$+ \sum_{k=0}^{K-1} \int_{\mathcal{X}_k} \left(\int_{\mathcal{X}_{k+1}} \lambda_{k+1}\left(y\right) \pi_k\left(y|x\right) dy - \lambda_k\left(x\right)\right) p_k\left(x\right) dx.$$

This is a separable problem in each $p_k$ and since $p_k$ is constrained to be a probability measure, the optimal choice of $p_k$ (for $k = 1, \dots, K-1$) is a $\delta$ measure with probability 1 assigned to the the $x \in \mathcal{X}_k$ that maximizes:

$$\int_{\mathcal{X}_{k+1}} \lambda_{k+1}\left(y\right) \pi_k\left(y|x\right) dy - \lambda_k\left(x\right).$$

The optimization over $p_0$ can be rewritten as follows:

$$\max_{p_0 \in \mathcal{P}_0} \int_{\mathcal{X}_0} \left(\int_{\mathcal{X}_1} \lambda_1\left(y\right) \pi_0\left(y|x\right) dy\right) p_0\left(x\right) dx.$$

The optimization over $p_K$ can be rewritten as follows:

$$\boldsymbol{\psi}^\star\left(\lambda_K\right) = \max_{p_K \in \mathcal{P}_K} \boldsymbol{\psi}\left(p_K\right) - \int \lambda_K\left(x\right) p_K\left(x\right) dx.$$

The overall dual problem can be rewritten as:

$$\boldsymbol{\psi}^\star\left(\lambda_K\right) + \sum_{k=1}^{K-1} \max_{x \in \mathcal{X}_k} \left(\int_{\mathcal{X}_{k+1}} \lambda_{k+1}\left(y\right) \pi_k\left(y|x\right) dy - \lambda_k\left(x\right)\right)$$

$$+ \max_{p_0 \in \mathcal{P}_0} \int_{\mathcal{X}_0} \left(\int_{\mathcal{X}_1} \lambda_1\left(y\right) \pi_0\left(y|x\right) dy\right) p_0\left(x\right) dx.$$

Writing this in terms of expected values, we obtain $g\left(\lambda\right)$. Plugging in $\lambda^\star$ into $g\left(\lambda\right)$, all the terms cancel except the first term which evaluates to:

$$\max_{p_0 \in \mathcal{P}_0} \mathbb{E}_{x \sim p_0} \left[\mathbb{E}_{y \in \pi_0(x)} \left[\lambda_1^\star\left(x\right)\right]\right] = \max_{p_0 \in \mathcal{P}_0} \mathbb{E}_{x \sim p_0} \left[\mathbb{E}_{y \in \pi_0(x)} \left[\mathbb{E}_{z \sim \pi_1(x)} \left[\lambda_2^\star\left(z\right)\right]\right]\right] \dots = \max_{p_0 \in \mathcal{P}_0} \boldsymbol{\psi}\left(p_0\right)$$

$\square$

## C   Additional Theoretical Results

### C.1   Computation of Expected Values

Since $\pi_k\left(x\right)$ is typically a distribution that one can sample from easily (as it is required to perform forward inference through the neural network), estimating this expectation via sampling is a viable option. However, in order to turn this into verified bounds on the specification, one needs to appeal to concentration inequalities and the final guarantees would only be probabilistically valid. We leave this direction for future work.

Instead, we focus on situations where the expectations can be computed in closed form. In particular, we consider layers of the form $\pi_k(x) = ws(x) + b, (w, b) \sim p_k^w$, where $s$ is an element-wise function like ReLU, sigmoid or tanh and $(w, b)$ represents a fully connected or convolutional layer. We consider a general form of Lagrange multipliers as a sum of quadratic and exponential terms as follows:

$$\lambda(x) = q^T x + \frac{1}{2} x^T Q x + \kappa \exp\left(\gamma^T x\right).$$

Let:

$$\tilde{s}(x) = \begin{pmatrix} 1 \\ s(x) \end{pmatrix}, \tilde{Q} = \begin{pmatrix} 0 & q^T \\ q & Q \end{pmatrix}, \tilde{w} = (b \quad w).$$

Then:

$$\lambda(Ws(x) + b) = \frac{1}{2}(\tilde{s}(x))^T \tilde{w}^T \tilde{Q} \tilde{w} \tilde{s}(x) + \kappa \exp\left(\gamma^T \tilde{w} \tilde{s}(x)\right).$$

Taking expected values with respect to $\tilde{W} \sim p_k^w$, we obtain:

$$\mathbb{E}\left[\lambda(ws(x) + b)\right] = \frac{1}{2}(\tilde{s}(x))^T \underset{\tilde{s}(x)}{\mathbb{E}}\left[\tilde{w}^T \tilde{Q} \tilde{w}\right] + \kappa \prod_{i,j} \mathbb{E}\left[\exp\left(\gamma_i \tilde{w}_{ij} \tilde{s}(x_j)\right)\right], \qquad (10)$$

where we have assumed that each element of $\tilde{W}_{ij}$ is independently distributed. The first term in (10) evaluates to:

$$\frac{1}{2}\text{Trace}\left(\text{Cov}\left(\tilde{w}\tilde{s}(x)\right)\tilde{Q}\right) + \frac{1}{2}\left(\mathbb{E}\left[\tilde{w}\tilde{s}(x)\right]\right)^\top \tilde{Q}\left(\mathbb{E}\left[\tilde{w}\tilde{s}(x)\right]\right),$$

and the second one to:

$$\kappa \prod_{i,j} \text{mgf}_{ij}\left(\gamma_i \tilde{s}_j(x)\right),$$

where $\text{Cov}(X)$ refers to the covariance matrix of the random vector, and $\text{mgf}_{ij}$ refers to the moment generating function of the random variable $\tilde{w}_{ij}$:

$$\text{mgf}_{ij}(\theta) = \mathbb{E}\left[\exp\left(\tilde{w}_{ij}\theta\right)\right].$$

The details of this computation for various distributions on $\tilde{w}$ (Gaussian posterior, MC-dropout) are worked out below.

**Diagonal Gaussian posterior.** Consider a BNN with a Gaussian posterior, $\tilde{w} \sim \mathcal{N}\left(\mu, \text{diag}\left(\sigma^2\right)\right)$, where $\mu, \sigma \in \mathbb{R}^{mn}$, let $\text{Mat}(\mu) \in \mathbb{R}^{m \times n}$ denote a reshaped version of $\mu$. Then, we have:

$$\mathbb{E}\left[\lambda(ws(x) + b)\right] = \frac{1}{2}\text{Trace}\left(\text{diag}\left(\text{Mat}\left(\sigma^2\right)\tilde{s}(x)\right)\tilde{Q}\right) + \frac{1}{2}(\text{Mat}(\mu)\tilde{s}(x))^T \tilde{Q}\left(\text{Mat}(\mu)\tilde{s}(x)\right)$$

$$+ \kappa \prod_{i,j} \exp\left(\text{Mat}(\mu)_{ij}\tilde{s}(x_j)\gamma_i + \frac{1}{2}\text{Mat}\left(\sigma^2\right)_{ij}\left(\tilde{s}(x_j)\gamma_i\right)^2\right).$$

**MC dropout.** Now assume a neural network with dropout: $\tilde{w} = \mu \odot \text{Bernoulli}(p_{\text{dropout}})$, where $\mu \in \mathbb{R}^{mn}$ denotes the weight in the absence of dropout and $p_{\text{dropout}} \in \mathbb{R}^{mn}$ denotes the probability of dropout. Then, we have:

$$\mathbb{E}\left[\lambda(ws(x) + b)\right] = \frac{1}{2}\text{Trace}\left(\text{diag}\left(\text{Mat}\left(\mu \odot p_{\text{dropout}} \odot (1 - p_{\text{dropout}})\right)\tilde{s}(x)\right)\tilde{Q}\right)$$

$$+ \frac{1}{2}(\text{Mat}\left(\mu \odot p_{\text{dropout}}\right)\tilde{s}(x))^T \tilde{Q}\left(\text{Mat}\left(\mu \odot p_{\text{dropout}}\right)\tilde{s}(x)\right)$$

$$+ \kappa \prod_{i,j}\left(\text{Mat}\left(p_{\text{dropout}}\right)_{ij}\exp\left(\text{Mat}(\mu)_{ij}\tilde{s}(x_j)\gamma_i\right) + 1 - \text{Mat}\left(p_{\text{dropout}}\right)_{ij}\right).$$

### C.2 Expected-Softmax Optimization

We describe an algorithm to solve optimization problems of the form

$$\max_{\ell \leq x \leq u} \mu^T \texttt{softmax}(x) - \lambda^T x$$

Our results will rely on the following lemma:

**Proposition 3.** *Consider the function*

$$f(x) = \frac{\sum_i \mu_i \exp(x_i) + D}{\sum_j \exp(x_j) + B} - \lambda^T x$$

*where $B \geq 0$ and $D = 0$ if $B = 0$. Let $r = \frac{D}{B}$ if $B > 0$ and $0$ otherwise. Define the set*

$$\Delta = \left\{ \kappa \in \mathbb{R} : (\kappa - r)\left(\prod_{i=1}^n (\mu_i - \kappa)\right) - \sum_{i=1}^n \mu_i (1 - r)\lambda_i \left(\prod_{j \neq i} (\mu_j - \kappa)\right) = 0 \right\}$$

*which is a set with at most $n + 1$ elements. Define further*

$$\Delta_f = \begin{cases} \left\{ \kappa \in \Delta : 0 < \frac{\lambda}{\mu - \kappa} < 1, \sum_{i=1}^n \frac{\lambda_i}{\mu_i - \kappa} \leq 1 \right\} & \text{if } B > 0 \\ \left\{ \kappa \in \Delta : 0 < \frac{\lambda}{\mu - \kappa} < 1, \sum_{i=1}^n \frac{\lambda_i}{\mu_i - \kappa} = 1 \right\} & \text{if } B = 0 \end{cases}$$

*Then, the set of stationary points of $f$ are given by*

$$\left\{ \log\left( h\left(\frac{\lambda}{\mu - \kappa}\right)\right) : \kappa \in \Delta_f \right\}$$

*where*

$$h(v) = \begin{cases} \frac{Bv}{1 - \mathbf{1}^T v} & \text{if } B > 0 \\ \{\theta v : \theta > 0\} & \text{if } B = 0 \end{cases}$$

*Proof.* Differentiating with respect to $x_i$, we obtain

$$\frac{\exp(x_i)}{\sum_j \exp(x_j) + B}\left(\mu_i - \left(\frac{\sum_j \mu_j \exp(x_j) + D}{\sum_j \exp(x_j) + B}\right)\right) - \lambda_i = p_i\left(\mu_i - \mu^T p - q\right) - \lambda_i$$

where

$$p_i = \frac{\exp(x_i)}{\sum_j \exp(x_j) + B}, q = \frac{D}{\sum_j \exp(x_j) + B}.$$

If we set the derivative to $0$ (to obtain a stationary point) we obtain the following coupled set of equations in $p, q, \kappa$:

$$p_i = \frac{\lambda_i}{\mu_i - \kappa} \quad i = 1, \ldots, n$$

$$q = r\left(1 - \sum_i p_i\right),$$

$$\kappa = \sum_i \mu_i p_i + q,$$

where $r = \frac{D}{B}$. We can solve this by first solving the scalar equation

$$\kappa - r = \sum_i \frac{\mu_i (1 - r)\lambda_i}{\mu_i - \kappa}$$

for $\kappa$ (this is derived by adding up the first $n$ equations above weighted by $\mu_i$ and plugging in the value of $q$). This equation can be converted into a polynomial equation in $\kappa$

$$(\kappa - r)\left(\prod_i (\mu_i - \kappa)\right) - \sum_i \mu_i (1 - r)\lambda_i \left(\prod_{j \neq i} (\mu_j - \kappa)\right) = 0$$

which we can solve for all possible real solutions, denote this set $\Delta$. Note that this set has at most $n + 1$ elements since it is the set of real solutions to a degree $n + 1$ polynomial.

In order to recover $x$ from this, we first recall:

$$p_i = \frac{\lambda_i}{\mu_i - \kappa}$$

Since $p_i = \frac{\exp x_i}{\sum_j \exp(x_j) + B}$, we require that $p_i \in (0, 1)$ and $\sum_i p_i \leq 1$ (with equality when $B = 0$ and strict inequality when $B = 1$. We thus filter $\Delta$ to the set of $\kappa$ that lead to $p$ satisfying these properties to obtain $\Delta_f$.

Once we have these, we are guaranteed that for each $\kappa \in \Delta_f$, we can define $p_i$ as above and solve for $x_i$ by solving the linear system of equations

$$u_i = p_i \left( \sum_j u_j + B \right) \quad i = 1, 2, \ldots, n$$

which can be solved as:

$$u = B \left( I - p\mathbf{1}^T \right)^{-1} p = \frac{Bp}{1 - \mathbf{1}^T p}, x = \log(u)$$

if $B > 0$ since the matrix $I - p\mathbf{1}^T$ is strictly diagonally dominant and hence invertible, and we applied the Woodbury identity to compute the explicit inverse.

If $B = 0$, we have $p = \texttt{softmax}(x)$ and can recover $x$ as

$$x = \log(p\theta)$$

for any $\theta > 0$. $\qquad \square$

The above lemma allows us to characterize all stationary points of the function

$$\mu^T \texttt{softmax}(x) - \lambda^T x$$

when a subset of entries of $x$ are fixed to their upper or lower bounds, and we search for stationary points wrt the remaining free variables. Given this ability, we can develop an algorithm to globally optimize $\mu^T \texttt{softmax}(x) - \lambda^T x$ subject to bound constraints by iterating over all possible $3^n$ configurations of binding constraints (each variable could be at its lower bound, upper bound or strictly between them). In this way, we are guaranteed to loop over all local optima, and by picking the one achieving the best objective value, we can guarantee that we have obtained the global optimum. The overall algorithm is presented in Algorithm 2.

**Proposition 4.** *Algorithm 2 finds the global optimum of the optimization problem*

$$\min_{x: \ell \leq x \leq u} \mu^T \texttt{softmax}(x) - \lambda^T x$$

*and runs in time $O(n3^n)$ where $n$ is the dimension of $x$.*

**Algorithm 2** Solving softmax layer problem via exhaustive enumeration

---

Inputs: $\lambda, \mu, \ell, u \in \mathbb{R}^n$
$x^\star \leftarrow \ell$
$f_{opt}(x) \leftarrow \mu^T \texttt{softmax}(x) - \lambda^T x, \ f^\star \leftarrow f_{opt}(x^\star)$
**for** $v \in [\text{Lower}, \text{Upper}, \text{Interior}]^n$ **do**

$\quad$ nonbinding$[i] \leftarrow (v[i] == \text{Interior}), \ x_i \leftarrow \begin{cases} l[i] \text{ if } v[i] = \text{Lower} \\ u[i] \text{ if } v[i] = \text{Upper} \end{cases} \quad$ for $i = 1, \ldots, n$

$$B \leftarrow \sum_{i \text{ such that nonbinding}[i]==\text{False}} \exp(x[i])$$

$$D \leftarrow \sum_{i \text{ such that nonbinding}[i]==\text{False}} \mu[i] \exp(x[i])$$

$\quad$ Use proposition 3 to find the set of stationary points $\mathcal{S}_f$ of the function

$$f(x) = \frac{\sum\limits_{i \text{ such that nonbinding}[i]} \mu_i \exp(x_i) + D}{\sum\limits_{j \text{ such that nonbinding}[j]} \exp(x_j) + B} - \sum_{j \text{ such that nonbinding}[j]} \lambda[j] x_j$$

$\quad$ **for** $x^s \in \mathcal{S}_f$ **do**
$\quad\quad$ **if** $x_i^s \in [\ell[i], u[i]] \quad \forall i$ s.t nonbinding$[i]$ **then**
$\quad\quad\quad$ $x_i \leftarrow x_i^s \quad \forall i$ s.t nonbinding$[i]$.
$\quad\quad\quad$ **if** $f_{opt}(x) > f^\star$ **then**
$\quad\quad\quad\quad$ $x^\star \leftarrow x$
$\quad\quad\quad\quad$ $f^\star \leftarrow f_{opt}(x^\star)$
$\quad\quad\quad$ **end if**
$\quad\quad$ **end if**
$\quad$ **end for**
**end for**
Return $x^\star, f^\star$

---

## C.3  Input Layer with Linear-Exponential Multipliers

We recall the setting from Proposition 1. Let $\lambda_1(x) = \alpha^T x + \exp(\gamma^T x + \kappa)$, $\lambda_2(x) = \beta^T x$, $g_0^\star = \max_{p_0 \in \mathcal{P}_0} g_0(p_0, \lambda_1)$, and $g_1^\star = \max_{x \in \mathcal{X}_1} g_0(x, \lambda_1, \lambda_2)$.

**Proposition 5.** *In the setting described above, and with $s$ as the element-wise activation function:*

$$g_0^\star \leq \alpha^T(w\mu + b) + \exp\left(\frac{\|w^T \gamma\|^2 \sigma^2}{2} + \gamma^T b + \kappa\right),$$

$$g_1^\star \leq \max_{\substack{x \in \mathcal{X}_2 \\ z = s(x)}} \beta^T(w_2 z + b_2) - \alpha^T x - \exp(\gamma^T x + \kappa).$$

*The maximization in the second equation can be bounded by solving the following convex optimization problem:*

$$\min_{\eta \in \mathbb{R}^n, \zeta \in \mathbb{R}_+} \zeta(\log(\zeta) - 1 - \kappa) + \mathbf{1}^T \max\left((\eta + w_2^T \beta) \odot s(l_2), (\eta + w_2^T \beta) \odot s(u_2)\right)$$

$$+ \sum_i s^\star(\alpha_i + \zeta\gamma_i, \eta_i, l_{2i}, u_{2i}),$$

*where $s^\star(a, b, c, d) = \max_{z \in [c,d]} -az - bs(z)$.*

*Proof.*

$$\max_{\substack{x \in \mathcal{X}_2 \\ z = s(x)}} \beta^T \left(w_2 z + b_2\right) - \alpha^T x - \exp\left(\alpha^T x + \kappa\right),$$

$$\leq \min_{\eta} \max_{x \in \mathcal{X}_2, z \in s(\mathcal{X}_2)} \eta^T \left(z - s(x)\right) + \beta^T \left(w_2 z + b_2\right) - \alpha^T x - \exp\left(\gamma^T x + \kappa\right),$$

$$\leq \min_{\eta, \zeta} \max_{x \in \mathcal{X}_2, t} \mathbf{1}^T \max\left(\left(\eta + w_2^T \beta\right) \odot s\left(l_2\right), \left(\eta + w_2^T \beta\right) \odot s\left(u_2\right)\right) + \beta^T b_2 - \alpha^T x$$

$$- \eta^T s(x) - \exp(t) + \zeta\left(t - \gamma^T x - \kappa\right),$$

$$\leq \min_{\eta, \zeta} \zeta\left(\log\left(\zeta\right) - 1 - \kappa\right) + \mathbf{1}^T \max\left(\left(\eta + w_2^T \beta\right) \odot s\left(l_2\right), \left(\eta + w_2^T \beta\right) \odot s\left(u_2\right)\right)$$

$$+ \max_{l_2 \leq x \leq u_2} - \left(\alpha + \zeta\gamma\right)^T x - \eta^T s(x),$$

$$\leq \min_{\eta, \zeta} \zeta\left(\log\left(\zeta\right) - 1 - \kappa\right) + \mathbf{1}^T \max\left(\left(\eta + w_2^T \beta\right) \odot s\left(l_2\right), \left(\eta + w_2^T \beta\right) \odot s\left(u_2\right)\right)$$

$$+ \sum_i \max_{l_{2i} \leq x_i \leq u_{2i}} - \left(\alpha_i + \zeta\gamma_i\right) x_i - \eta_i s\left(x_i\right).$$

$\square$

## C.4  Inner Problem with Linear Multipliers

In its general form, the objective function of the inner maximization problem can be expressed as:

$$g_k(x_k, \lambda_k, \lambda_{k+1}) = \mathop{\mathbb{E}}_{y \sim \pi_k(x_k)} \left[\lambda_{k+1}\left(y\right)\right] - \lambda_k\left(x_k\right). \tag{11}$$

We assume that the layer is in the form $y = W s(x) + b$, where $W$ and $b$ are random variables and $s$ is an element-wise activation function. Then we can rewrite $g_k(x_k, \lambda_k, \lambda_{k+1})$ as:

$$g_k(x_k, \lambda_k, \lambda_{k+1}) = \mathop{\mathbb{E}}_{W, b} \left[\lambda_{k+1}\left(W \max\{x_k, 0\} + b\right)\right] - \lambda_k\left(x_k\right). \tag{12}$$

We now use the assumption that $\lambda_{k+1}$ is linear: $\lambda_{k+1} : y \mapsto \theta_{k+1}^\top y$. Then the problem can equivalently be written as:

$$
\begin{aligned}
g_k(x_k, \lambda_k, \lambda_{k+1}) &= \mathop{\mathbb{E}}_{W, b} \left[\theta_{k+1}^\top \left(W s\left(x_k\right) + b\right)\right] - \theta_k^T x_k, \\
&= \left(\mathbb{E}\left[W\right]^\top \theta_{k+1}\right)^\top s\left(x_k\right) + \theta_{k+1}^\top \mathbb{E}\left[b\right] - \theta_k^T x_k, \\
&= \theta_{k+1}^T \mathbb{E}\left[b\right] + \sum_i \left(\mathbb{E}\left[W\right]^\top \theta_{k+1}\right)_i s\left(x_i\right) - \left(\theta_k\right)_i x_i.
\end{aligned}
\tag{13}
$$

Maximizing the RHS subject to $l \leq x \leq u$, we obtain:

$$\theta_{k+1}^T \mathbb{E}\left[b\right] + \sum_i \max_{z \in [l_i, u_i]} \left(\mathbb{E}\left[W\right]^\top \theta_{k+1}\right)_i s\left(z\right) - \left(\theta_k\right)_i z.$$

where the maximization over $z$ can be solved in closed form for most common activation functions $s$ as shown in Dvijotham et al. [2018b].

So we can simply apply the deterministic algorithm to compute the closed-form solution of this problem.

## D  Relationship to Prior work

We establish connections between the functional Lagrangian framework and prior work on deterministic verification techniques based on Lagrangian relaxations and SDP relaxations.

### D.1 Lagrangian Dual Approach

We assume that the network layers are deterministic layers of the form:

$$\forall\, k, k \text{ is odd } \pi_k(x) = w_k x + b_k,$$
$$\forall\, k, k \text{ is even } \pi_k(x) = s(x), \tag{14}$$

where $s$ is an element-wise activation function and that the specification can be written as:

$$\psi(x_K) = c^T x_K. \tag{15}$$

**Proposition 6** (Linear Multipliers). *For a verification problem described by equations (14, 15), the functional Lagrangian framework with linear functional multipliers $\lambda_k(x) = \theta_k^T x$ is equivalent to the Lagrangian dual approach from Dvijotham et al. [2018b].*

*Proof.* The final layer problem is

$$\max_{x_K} c^T x_K - \theta_K^T x_K = \mathbf{1}^T \max\left((c - \theta_K) \odot l_K, (c - \theta_K) \odot u_K\right)$$

For even layers with $k < K$, the optimization problem is

$$\max_{x \in [l_k, u_k]} \theta_{k+1}^T (w_k x + b_k) - \theta_k^T x = \theta_{k+1}^T b_k + \left(w_k^T \theta_{k+1} - \theta_k\right)^T x$$
$$= \mathbf{1}^T \max\left(\left(w_k^T \theta_{k+1} - \theta_k\right) \odot l_k, \left(w_k^T \theta_{k+1} - \theta_k\right) \odot u_k\right) + \theta_{k+1}^T b_k$$

For odd layers with $k < K$, the optimization problem is

$$\max_{x \in [l_k, u_k]} \theta_{k+1}^T s(x) - \theta_k^T x = \sum_i \max_{z \in [l_{ki}, u_{ki}]} (\theta_{k+1})_i s(z) - (\theta_k)_i z$$

All these computations precisely match those from Dvijotham et al. [2018b], demonstrating the equivalence. $\qquad\square$

### D.2 SDP-cert

We assume that the network layers are deterministic layers of the form:

$$\forall\, k\ \pi_k(x) = \texttt{ReLU}(w_k x + b_k) \tag{16}$$

where $s$ is an element-wise activation function and that the specification can be written as:

$$\psi(x_K) = c^T x_K. \tag{17}$$

**Proposition 7** (Quadratic Multipliers). *For a verification problem described by equations (16, 17), the optimal value of the Functional Lagrangian Dual with*

$$\lambda_k(x) = q_k^T x + \frac{1}{2} x^T Q_k x \quad k = 1, \dots, K - 1$$
$$\lambda_K(x) = q_K^T x$$

*and when an SDP relaxation is used to upper bound the inner maximization problems over $g_k$, is equal to the dual of the SDP relaxation from Raghunathan et al. [2018].*

*Proof.* With quadratic multipliers of the form $\lambda_k(x) = q_k^T x + \frac{1}{2} x^T Q_k x \quad k = 1, \dots, K - 2$ and $\lambda_K(x) = q_K^T x$, the inner maximization problems for the intermediate layers are of the form:

$$\max_{\substack{x \in [l, u] \\ y = \texttt{ReLU}(wx + b)}} \tilde{q}^T y + \frac{1}{2} y^T \tilde{Q} y - q^T x - \frac{1}{2} x^T Q x,$$

where $l = l_k, u = u_k, \tilde{q} = q_{k+1}, \tilde{Q} = Q_{k+1}, q = q_k, Q = Q_k, w = w_k, b = b_k$. Let $x \in \mathbb{R}^n, y \in \mathbb{R}^m$ ($n$ dimensional input, $m$ dimensional output of the layer). Further, let $\tilde{l} = l_{k+1}, \tilde{u} = u_{k+1}$.

We can relax the above optimization problem to the following Semidefinite Program (SDP) (following Raghunathan et al. [2018]):

$$\max_P \tilde{q}^T P[y] + \frac{1}{2}\text{Trace}\left(\tilde{Q}P[yy^T]\right) - q^T P[x] - \frac{1}{2}\text{Trace}\left(QP[xx^T]\right) \tag{18a}$$

$$\text{Subject to } P = \begin{pmatrix} 1 & (P[y])^T & (P[x])^T \\ P[y] & P[yy^T] & P[xy^T] \\ P[x] & (P[xy^T])^T & P[xx^T] \end{pmatrix} \in \mathbb{S}^{n+m+1}, \tag{18b}$$

$$P \succeq 0, \tag{18c}$$

$$\text{diag}\left(P[xx^T] - l\left(P[x]\right)^T - P[x]u^T + lu^T\right) \leq 0, \tag{18d}$$

$$\text{diag}\left(P[yy^T] - \tilde{l}\left(P[y]\right)^T - P[y]\tilde{u}^T + \tilde{l}\tilde{u}^T\right) \leq 0, \tag{18e}$$

$$P[y] \geq 0, P[w] \geq wP[x], \tag{18f}$$

$$\text{diag}\left(wP[xy^T]\right) + P[y] \odot b = \text{diag}\left(P[yy^T]\right). \tag{18g}$$

where the final constraint follows from the observation that $y \odot (y - wx - b) = 0$.

The above optimization problem resembles the formulation of Raghunathan et al. [2018] except that it only involves two adjacent layers rather than all the layers at once. Let $\Delta_k$ denote the feasible set given the constraints in the above optimization problem. Then, the formulation of Raghunathan et al. [2018] can be written as:

$$\max \ c^T y_K \tag{19a}$$

$$\text{subject to } P_k \in \Delta_k \quad k = 0, \dots, K-1, \ l_K \leq y_K \leq u_K, \tag{19b}$$

$$P_{k+1}[xx^T] = P_k[yy^T] \quad k = 0, \dots, K-2, \tag{19c}$$

$$P_{k+1}[x] = P_k[y] \quad k = 0, \dots, K-2, \tag{19d}$$

$$y_K = P_{K-1}[y]. \tag{19e}$$

Note that in Raghunathan et al. [2018], a single large $P$ matrix is used whose block-diagonal sub-blocks are $P_k$ and the constraint $P \succeq 0$ is enforced. Due to the matrix completion theorem for SDPs [Grone et al., 1984, Vandenberghe and Andersen, 2015], it suffices to ensure postitive semidefiniteness of the sub-blocks rather than the full $P$ matrix.

Dualizing the last three sets of constraints above with Lagrangian multipliers $\Theta_k \in \mathbb{S}^{n_k+n_{k+1}+1}, \theta_k \in \mathbb{R}^{n_k}$ and $\theta_K \in \mathbb{R}^{n_K}$, we obtain the following optimization problem:

$$\max \ c^T y_K + \theta_K^T\left(-P_{K-1}[y] + y_K\right) + \sum_{k=0}^{K-2}\text{Trace}\left(\Theta_k\left(P_{k+1}[xx^T] - P_k[yy^T]\right)\right)$$

$$+ \sum_{k=0}^{K-2}\theta_k^T\left(P_{k+1}[x] - P_k[y]\right)$$

$$\text{subject to } P_k \in \Delta_k \quad k = 0, \dots, K-1,$$

$$l_K \leq y_K \leq u_K.$$

The objective decomposes over $P_k$ and can be rewritten as:

$$\max_{l_K \leq y_K \leq u_K}(c + \theta_K)^T y_K + \sum_{k=0}^{K-1}\max_{P_k \in \Delta_k}\text{Trace}\left(\Theta_{k-1}P_k[xx^T]\right) + \theta_{k-1}^T P_k[x] - \theta_k^T P_k[y]$$

$$- \text{Trace}\left(\Theta_k P_k[yy^T]\right), \tag{20}$$

with the convention that $\Theta_{K-1} = 0, \Theta_{-1} = 0, \theta_{-1} = 0$. If we set $Q_k = -\Theta_{k-1}, q_k = -\theta_{k-1}$ for $k = 1, \dots, K$, then the optimization over $P_k$ precisely matches the optimization in (18). Further, since $\lambda_K$ is linear, the final layer optimization simply reduces to:

$$\max_{l_K \leq x_K \leq u_K} c^T x_K - q_K^T x_K,$$

which matches the first term in (20).

Thus, the functional Lagrangian framework with quadratic multipliers $\lambda_k$ for $k = 1, \ldots, K - 2$ and a linear multiplier for $\lambda_K$ precisely matches the Lagrangian dual of (19) and since (19) is a convex optimization problem, strong duality guarantees that the optimal values must coincide.

$\square$

# E  Additional Experimental Details

## E.1  Robust OOD Detection on Stochastic Neural Networks

**Inner Optimization.**  All inner problems have a closed-form as shown in section C.4, except for the last one, which is handled as follows.

The last inner problem can be formulated as:

$$\max_{x_K \in \mathcal{X}_K} \mu^\top \texttt{softmax}(x_K) + \nu^\top x_K, \tag{21}$$

where $\mu$ is a one-hot encoded vector and $\nu$ is a real-valued vector.

- Projected Gradient Ascent (Training):
    - Hyper-parameters: we use the Adam optimizer Kingma and Ba [2015], with a learning-rate of 1.0 and a maximum of 1000 iterations.
    - Stopping criterion: when all coordinates have either zero gradient, or are at a boundary with the gradient pointing outwards of the feasible set.
    - In order to help the gradient method find the global maximum, we use a heuristic for initialization, which consists of using the following two starting points for the maximization (and then to take the best of the corresponding two solutions found):
        1. Ignore affine part ($\nu = 0$), which gives a solution in closed form: set $x_K$ at its upper bound at the coordinate where $\mu$ is 1, and at its lower bound elsewhere.
        2. Ignore softmax part ($\mu = 0$), which also gives a solution in closed form: set $x_K$ at its upper bound at the coordinates where $\nu \geq 0$, and at its lower bound elsewhere.
- Evaluation: we use Algorithm 2 at evaluation time, which solves the maximization exactly.

**Outer Optimization.**  We use the Adam optimizer, with a learning-rate that is initialized at 0.001 and divided by 10 every 250 steps. We run the optimization for a total of 1000 steps.

**Gaussian-MLP.**  We use the ReLU MLP from [Wicker et al., 2020] that consists of 2 hidden layers of 128 units each. The models are available at `https://github.com/matthewwicker/ProbabilisticSafetyforBNNs`.

**LeNet.**  We use the LeNet5 architecture with dropout applied to the last fully connected layer with a probability of 0.5. To make the bound-propagation simpler, we do not use max-pooling layers and instead increase the stride of convolutions.

**VGG-X.**  For VGG-X (where $X \in \{2, 4, 8, 16, 32, 64\}$), the architecture can be described as:

- Conv 3x3, X filters, stride 1
- ReLU
- Conv 3x3, X filters, stride 2
- ReLU
- Conv 3x3, 2X filters, stride 2
- ReLU
- Conv 3x3, 2X filters, stride 2
- ReLU
- Flatten

- Linear with 128 output neurons
- Dropout with rate 0.2
- Linear with 10 output neurons

**Hardware**   The verification of each sample is run on a CPU with 1-2 cores (and on each instance, BP and FL are timed on the same exact hardware configuration).

### E.2   Adversarial Robustness for Stochastic Neural Networks

**Inner Optimization.**   We use a similar approach as in Appendix E.1. For the final inner problem (corresponding to the objective which is a linear function of the softmax and the layer inputs), we run projected gradient ascent during the optimization phase and then use Algorithm 2 to solve the maximization exactly. For projected gradient ascent, because of the non-convexity of the problem, we use the following heuristics to try and find the global maximum:

- Black-box attack (1st phase): we use the `Square` adversarial attack Andriushchenko et al. [2020], with 600 iterations, 300 random restarts and learning-rate of 0.1.
- Fine-tuning (2nd phase): We then choose the best attack from the restarts, and employ projected gradient ascent, with a learning-rate of 0.1 and 100 iterations to fine-tune further.

**Model Parameters.**   We use the 1 and 2 layer ReLU MLPs from [Wicker et al., 2020]. The models are available at `https://github.com/matthewwicker/ProbabilisticSafetyforBNNs`.

**Outer Optimization.**   We use the Adam optimizer, with a learning-rate that is initialized at 0.001 and divided by 10 every 1000 steps. We run the optimization for a total of 3000 steps.

**Hardware**   All experiments were run on a P100 GPU.

### E.3   Distributionally Robust OOD Detection

**Model.**   We train networks on MNIST using the code from `https://gitlab.com/Bitterwolf/GOOD` with the CEDA method, and with the default hyperparameters. We train a CNN with `ReLU` activations the following layers:

- Conv 4x4, 16 filters, stride 2, padding 2 on both sides
- ReLU
- Conv 4x4, 32 filters, stride 1, padding 1 on both sides
- Relu
- Flatten
- Linear with 100 output neurons
- Relu
- Linear with 10 output neurons

**Outer Optimization.**   For the outer loop of the verification procedure, we use Adam for 100k steps. The learning-rate is initially set to 0.0001 and then divided by 10 after 60k and 80k steps.

**Hardware**   We run the experiments for this section on a CPU with 2-4 cores.

## F   Additional Results with Interval Bound Propagation for Bilinear Operations

### F.1   Robust OOD Detection for Stochastic Neural Networks

We repeat the experiments in Section 5.1 where we use IBP to handle bound-propagation through the layers where bilinear propagation is required (because of bounds coming from both the layer

Table 4: Robust OOD Detection: MNIST vs EMNIST (MLP and LeNet) and CIFAR-10 vs CIFAR-100 (VGG-*). BP: Bound-Propagation (baseline), using IBP instead of Bunel et al. [2020] for bilinear operations; FL: Functional Lagrangian (ours). The reported times correspond to the median of the 500 samples.

| OOD Task | Model | #neurons | #params | $\epsilon$ | Time (s) | | GAUC (%) | | AAUC (%) |
|---|---|---|---|---|---|---|---|---|---|
| | | | | | BP | FL | BP | FL | |
| (E)MNIST | MLP | 256 | 2k | 0.01 | 1.1 | +13.1 | 55.4 | **67.5** | 86.9 |
| | | | | 0.03 | 1.2 | +13.4 | 38.7 | **54.5** | 88.6 |
| | | | | 0.05 | 1.3 | +17.7 | 19.1 | **36.0** | 88.8 |
| (E)MNIST | LeNet | 0.3M | 0.1M | 0.01 | 50.1 | +13.1 | 0.0 | **28.4** | 71.6 |
| | | | | 0.03 | 54.7 | +13.7 | 0.0 | **11.7** | 57.6 |
| | | | | 0.05 | 79.4 | +24.8 | 0.0 | **2.3** | 44.0 |
| CIFAR | VGG-16 | 3.0M | 83k | 0.001 | 426.4 | +21.4 | 0.0 | **21.7** | 60.9 |
| | VGG-32 | 5.9M | 0.2M | 0.001 | 1035.2 | +21.3 | 0.0 | **23.8** | 64.7 |
| | VGG-64 | 11.8M | 0.5M | 0.001 | 8549.7 | +42.1 | 0.0 | **28.6** | 67.4 |

Table 5: Adversarial Robustness for different BNN architectures trained on MNIST from Wicker et al. [2020]. BP: Bound-Propagation (baseline), using IBP instead of LBP for bilinear operations; FL: Functional Lagrangian (ours). The accuracy reported for FL and BP is the % of samples we can certify as robust with probability 1. For each model, we report results for the first 500 test-set samples.

| #layers | $\epsilon$ | #neurons | BP Acc. (%) | FL Acc. (%) | BP Time (s) | FL Time (s) | Adv Acc (%) |
|---|---|---|---|---|---|---|---|
| 1 | 0.025 | 128 | 43.8 | **65.2** | 1.3 | +353.3 | 82.6 |
| | | 256 | 40.6 | **64.6** | 1.4 | +431.3 | 82.6 |
| | | 512 | 35.0 | **57.0** | 1.3 | +357.1 | 82.8 |
| 2 | 0.001 | 256 | 29.4 | **36.9** | 1.6 | +439.6 | 79.4 |
| | | 512 | 46.0 | **63.4** | 1.7 | +433.8 | 89.2 |
| | | 1024 | 18.4 | **19.6** | 1.6 | +440.9 | 74.8 |

inputs and the layer parameters due to the stochasticity of the model) instead of Bunel et al. [2020]. Bunel et al. [2020] usually results in significantly tighter bounds compared to IBP but we note that for MNIST-CNN and CIFAR-CNN, we expect IBP to perform competitively as the bilinear bound propagation is only applied for a single layer (dropout). The results are presented in Table 4, and we find that even while using IBP as the bound-propagation method, our framework provides significantly stronger guarantees.

## F.2  Adversarial Robustness for Stochastic Neural Networks

For the verification tasks considered in Section 5.2, we use IBP instead of the tighter LBP as the bound-propagation method and report results in Table 2. We find that, similar to Section 5.2, our framework is able to significantly improve on the guarantees the bound-propagation baseline is able to provide.