# OpenReview forum: "Make Sure You're Unsure: A Framework for Verifying Probabilistic Specifications"
_NeurIPS.cc/2021/Conference — NeurIPS 2021 Spotlight_

### Official Review · Reviewer_XbVP · 2021-07-13

**Rating:** 7
**Confidence:** 3

**Summary:**

The paper proposes a framework for giving mathematical guarantees on neural networks with probabilistic specifications, e.g. for robustness across distributions or for robustness of models with non-deterministic outputs. The authors first formulate the abstract framework in terms of functional lagrangians and then show how to apply these by using concrete applications and concrete function classes for the lagrangians.

**Limitations And Societal Impact:**

I think the authors adequately address this.

**Main Review:**

Strengths:
The paper tackles an important problem and provides general framework from which further methods can be developed. The novel task of worst-case robustness with respect to a family of out-distributions is a valuable contribution. A fair number of examples for the application of the authors' proposed method is given. Implementing the method is technically non-trivial and thus a contribution (more so if the code is published).

Weaknesses:
As the authors openly admit, their approach can only provide certificates for models with outputs that are bounded with probability 1 in each layer, which does pose a practical limitation.
It is not always clear from the experiments how tight the final bounds are. Table 1 computes an AAUC as upper bound on the worst-case AUC but in Table 3 it should also be possible to give an empirical upper bound on the worst-case AUC by empirically finding an adversarial noise distribution that produces a low AUC.
Because the models were trained in a certification agnostic way, some of the threat models (CIFAR in Table 1 and 2 layers in Table 2) are so small that they would not even allow for any change at all when pixel values are constrained to lie on i/255 for i in {0,...,255}. As such, these guarantees seem fairly meaningless in practice. Using an epsilon of at least 1/255 would help fix this.

Correctness:
As far as I verified (no proofs) the math seems correct.

Clarity:
It seems to me that line 109 has sloppy notation when defining distributionally robust OOD detection. omega is in the set of distributions, but $x_\mathrm{ood}$ is in $\mathbb{R}^d$. It is clear what the authors meant to convey but this should be improved.
While the example starting in line 164 is simple and illustrative, I don't find it obvious how to actually write this in terms of probabilistic neural network layers as defined in Section 2.1. I think writing this explicitly (what is $\pi(x_1|x_0)$ etc.) in the appendix would help make the paper more readable.

Relation to prior work:
The relation to prior work is sufficiently discussed.

Reproducibility:
Unfortunately, the authors did not provide code. However, the information in the appendix seems sufficient in order to implement their method.

Additional feedback:
Formatting in Table headers. Shouldn't be all italic.
Line 47: "for e.g." = "for for example"
Formatting of "softmax" in line 234 is inconsistent with formatting in lines 110, 118, 232.
Inconsistent capitalization of "Section" between main text and checklist.

**Time Spent Reviewing:**

6

---

> ### Author Response · Authors · 2021-08-10
> **Thanks for the Comments. Response.**
>
> We thank the reviewer for their detailed comments. We answer point by point below:
>
> **Q:** It is not always clear from the experiments how tight the final bounds are. Table 1 computes an AAUC as upper bound on the worst-case AUC but in Table 3 it should also be possible to give an empirical upper bound on the worst-case AUC by empirically finding an adversarial noise distribution that produces a low AUC.
>
> **A:** We have included results with regard to the empirical worst-case performance for Table 2 (please see official comment above).  For Table 3, a meaningful empirical bound is difficult to obtain since the feasible set $\mathcal{P}_0$  is a non-parametric set of probability distributions, which makes it challenging to sample from or optimize over. We would be happy to consider any suggestions the reviewer may have regarding empirically finding an adversarial noise distribution.
>
> ---
>
> **Q:** Because the models were trained in a certification agnostic way, some of the threat models (CIFAR in Table 1 and 2 layers in Table 2) are so small that they would not even allow for any change at all when pixel values are constrained to lie on i/255 for i in {0,...,255}. As such, these guarantees seem fairly meaningless in practice. Using an epsilon of at least 1/255 would help fix this.
> We point out that the values we used in this work are the same as in Wicker et al. 2021.
>
> **A:** We agree that it would be interesting to explore the regime where epsilon is larger, which we expect to become possible through (1) better bound-propagation techniques, and / or (2) adversarial or verified training of the neural network (for instance using Wicker et al, 2021 for verified training of Bayesian neural networks).
>
> ---
> **Q:** It seems to me that line 109 has sloppy notation when defining distributionally robust OOD detection. omega is in the set of distributions, but xood is in Rd. It is clear what the authors meant to convey but this should be improved.
>
> **A:** We are replacing $x_\text{ood}$ with $\delta_{x_\text{ood}}$ (Dirac distribution around $x_\text{ood}$) to make the notation more precise.
>
> ---
> **Q:** While the example starting in line 164 is simple and illustrative, I don't find it obvious how to actually write this in terms of probabilistic neural network layers as defined in Section 2.1. I think writing this explicitly (what is π(x1|x0) etc.) in the appendix would help make the paper more readable.
>
> **A:** The reviewer is correct that it was not obvious how to write the example with the formalism previously described in the submission. We will update the submission with the example below to make this more transparent. Thanks for bringing this to our attention.
>
> Updated example:
> Let $\mathcal{P}\_0$ be the set of probability distributions with mean $0$, variance $1$, and support $[-1, 1]$, and let $\mathcal{N}\_{[a, b]}(\mu, \sigma^2)$ denote the normal distribution with mean $\mu$ and variance $\sigma^2$ with truncated support $[a, b]$. Now consider the following problem, for which we would like to compute an upper bound:
> $$
>  \texttt{OPT} = \max_{p_0 \in \mathcal{P}\_0} \mathbb{E}\_{X\_1}[\exp(-X\_1)]  \quad \text{s.t. } X_1|X_0 \sim \mathcal{N}_{[0, 1]}(X_0^2, 1) \text{ and } X_0 \sim p_0.
> $$
>
> This problem has two difficulties that prevent us from applying traditional optimization approaches like Lagrangian duality [Bertsekas, 2015], which has been used in neural network verification [Dvijotham et al., 2018]. The first difficulty is that the constraint linking $X_1$ to $X_0$ is stochastic, and standard approaches can not readily handle that. Second, the optimization variable $p_0$ can take any value in an entire set of probability distributions, while usual methods can only search over sets of real values. Thus standard methods fail to provide the tools to solve such a problem. Since the probability distributions have bounded support, a possible way around this problem is to ignore the stochasticity of the problem, and to optimize over the worst-case realization of the random variable $X_1$ in order to obtain a valid upper bound on $\texttt{OPT}$:
> $$
> \texttt{OPT} \leq \max_{x_1 \in [0, 1]} \exp(-x_1)  = 1.
> $$
> However this is an over-pessimistic modeling of the problem and thus the resulting upper bound is loose. In contrast, our main result (Theorem 1) shows that for any function $\lambda: \mathbb{R} \to \mathbb{R}$, $\texttt{OPT}$ can be upper bounded by:
> $$
> \texttt{OPT}
> \leq \max\_{x\_1 \in [0, 1], p\_0 \in \mathcal{P}\_0} \exp(-x\_1) - \lambda(x\_1)  + \mathbb{E}_{X\_0 \sim p\_0} [\mathbb{E}\_{X\_1|X\_0 \sim \mathcal{N}\_{[0, 1]}(X_0^2, 1)}[\lambda(X_1)]].
> $$
>
> This inequality holds true in particular for any function $\lambda$ of the form $x \mapsto \theta x$ where $\theta \in \mathbb{R}$, and thus:
> $$
> \begin{align*}
> \texttt{OPT}
> &\leq \inf_{\theta \in \mathbb{R}} \max_{x_1 \in [0, 1], p_0 \in \mathcal{P}\_0} \exp(-x_1) - \theta x_1  + \mathbb{E}\_{X_0 \sim p_0} [\mathbb{E}\_{X_1|X_0 \sim \mathcal{N}\_{[0, 1]}(X_0^2, 1)}[\theta X_1 ]], \\newline
> &= \inf_{\theta \in \mathbb{R}} \max_{x_1 \in [0, 1], p_0 \in \mathcal{P}\_0} \exp(-x_1) - \theta x_1  + \theta \mathbb{E}\_{X_0 \sim p_0} [X_0^2], \\newline
> &= \inf\_{\theta \in \mathbb{R}} \max_{x_1 \in [0, 1]} \exp(-x_1) - \theta x_1 + \theta  \approx 0.37.
> \end{align*}
> $$
>
> In other words, our framework has allowed us to derive a tractable upper bound on $\texttt{OPT}$ despite the difficulty of the original problem. This upper bound is significantly tighter than naive support-based bounds because our framework is able to capture much more structure.
>
> ---
> **Q:** Implementing the method is technically non-trivial and thus a contribution (more so if the code is published). Reproducibility: Unfortunately, the authors did not provide code.
>
> **A:** We commit to make the source code publicly available and ensure that the experiments are reproducible.
>
> ---
> **Q:** Formatting in Table headers. Shouldn't be all italic. Line 47: "for e.g." = "for for example" Formatting of "softmax" in line 234 is inconsistent with formatting in lines 110, 118, 232. Inconsistent capitalization of "Section" between main text and checklist.
>
> **A:** Thanks for bringing these points to our attention, we have fixed all of these issues for future versions of the paper (except for the softmax formatting l. 234, which is simply the italicized version of the same latex command used in the other places).
>
> ---

---

> > ### Comment · Reviewer_XbVP · 2021-09-10
> > **Respone**
> >
> > I thank the authors for their thorough response and for reworking the parts of the paper that I questioned.
> > My score remains unchanged.

---

### Official Review · Reviewer_jAE7 · 2021-07-17

**Rating:** 6
**Confidence:** 3

**Summary:**

This paper proposes a general framework for probabilistic robustness verification, with both stochastic networks and probabilistic specifications. The core technique is based on the generalization of Lagrangian duality where the multipliers are functional multipliers and probabilistic constraints are allowed. Under the proposed framework, the paper demonstrates several applications, including robust OOD detection on stochastic NN, adversarial robustness verification for stochastic NN, and distributionally robust OOD detection.

**Limitations And Societal Impact:**

The paper has discussed them in the last section.

**Main Review:**

Pros:
* The framework this paper proposes is general one for probabilistic robustness verification, that supports both stochastic networks and probabilistic specifications. And the verification has certified guarantees.
* There are empirical demonstrations for the proposed framework, which show the benefit of using functional Lagrangian.
* The paper is clearly written on introducing the techniques, experiments, limitations and social impact.

Cons:
* The proposed method seems slow (much slower than baseline).


**Time Spent Reviewing:**

2

---

> ### Author Response · Authors · 2021-08-10
> **Thanks for the Feedback. Response to Concern regarding Algorithm Efficiency**
>
> We thank the reviewer for their feedback. We answer their main concern below:
>
> **Concern:** The proposed method seems slow (much slower than baseline).
>
> **Response:** For any fixed runtime budget, our method gives a tighter bound than the baselines as i) the bounds from our approach are anytime, and 2) our initialization scheme recovers the bounds from the baseline (IBP). The anytime property of our algorithm allows us to trade off run-time for tighter bounds.
>
> We also note that in our experiments, we focus on obtaining the tightest bounds possible (for the aforementioned choices of lagrangian functionals) given a feasible compute budget, and tune the hyperparameters with this objective (as opposed to tuning for speed and tightness jointly, or re-tuning at many different compute budgets).
>
>
> While the much faster and simple baseline based on interval-bound propagation often provides weak and vacuous guarantees (Table 1), our method is able to significantly improve on this at non-prohibitive compute costs, while scaling to networks of the scale of VGG-64 with 11.8 million activations. However, we agree that carefully studying the optimization landscape of the parameters of the functional lagrangian, and developing efficient optimization algorithms is an interesting direction for further improving and scaling our method.

---

### Official Review · Reviewer_361J · 2021-07-21

**Rating:** 6
**Confidence:** 3

**Summary:**

This paper presents a general formulation of probabilistic specifications for neural networks which can capture both probabilistic networks and uncertain inputs. The authors generalize the notion of Lagrangian duality by defining functional Lagrange multipliers that can be arbitrary functions of the activations at a given layer.

**Main Review:**

Strengths
------------
- Probabilistic specifications are an important yet underexplored area in neural network verification
- The proposed general framework can handle stochasticity in both the specifications and the network architectures in a unified manner
- The functional Lagrangian formulation is interesting
- The results demonstrate gains over existing probabilistic verification methods for some properties

Weaknesses
-----------------
- No clear demonstration of the relationship between the probabilistic guarantees and actual attack success rate using SOTA attacks.
- It is not clear whether it is feasible for the framework to verify other types of property beyond the ones presented in the paper.

The authors propose a new probabilistic verification framework for neural networks that can handle stochasticity both in the input specifications as well as model architectures. The proposed functional Lagrangian formulation is interesting. The results seem to indicate gains over existing probabilistic verification methods. However, the implications of probabilistic verification in terms of attack success are not clear to me. What are the actual attack success rates for probabilistically verified networks? Are they lower than unverified ones? The authors did not provide any data. Moreover, it is not clear how generic the proposed framework is. The authors mention that in many cases obtaining a non-trivial upper bound is challenging. What are such cases? Beyond the properties presented in the paper, what other types of properties can be supported by the framework?


**Time Spent Reviewing:**

2

---

> ### Author Response · Authors · 2021-08-10
> **Thanks for the Comments. Response.**
>
> We thank the reviewer for their detailed comments. We answer point by point below:
>
> **Q:** No clear demonstration of the relationship between the probabilistic guarantees and actual attack success rate using SOTA attacks.
>
> **A:** Our framework is guaranteed to provide valid upper bounds on the attack success rate using *any* attack, and we note that the guarantee itself is not probabilistic, while the specification can have probabilistic components to it. The guarantee we are able to provide is independent of the attack.
>
> In Table 1, we provide empirical estimates of the worst-case AUC (AAUC) as determined by an adversarial attack based on sampling weights from the BNN weight-distribution, and we see that our framework significantly bridges the gap between prior work and the estimated worst-case AUC. We have included similar results for comparison with an actual adversarial attack for Table 2 as well, finding that our method again bridges a lot of the gap between the verified guarantees and the empirical estimates. In both our settings, our framework provides a valid bound on the performance as guaranteed by the theory.
>
> ---
>
> **Q:**  What are the actual attack success rates for probabilistically verified networks? Are they lower than unverified ones? The authors did not provide any data.
>
> **A:** We would like to highlight that the probabilistically verified networks are the exact same networks as the unverified ones. What differs between the metrics is the evaluation procedure they have been subjected to.
>
> In Table 1, the AAUC is an upper bound on the robustness, an empirical estimate obtained by adversarial attacks which may or may not have found the optimal adversary. On the other hand, the GAUC computed by our verification procedure is a  guaranteed lower bound on this robustness measure for the exact same network (for any attack).
>
> ---
>
> **Q:** It is not clear whether it is feasible for the framework to verify other types of property beyond the ones presented in the paper. Moreover, it is not clear how generic the proposed framework is. The authors mention that in many cases obtaining a non-trivial upper bound is challenging. What are such cases?
>
> **A:** In theory, a wide range of probabilistic specifications fit within our framework. We refer the reviewer to  Appendix A for additional examples that complement those of section 2.3. We also re-emphasize that for the specifications described within the paper, we empirically demonstrate that our framework is able to overcome the optimization challenges and significantly improve over the verification performance of existing approaches.
>
> Independent of the specification being studied, arbitrary complex Lagrangian multipliers (e.g., multi-layer perceptrons or polynomials) can yield intractable sub-problems that are challenging to obtain tight upper bounds for, and this is a natural consequence of the framework becoming exactly tight arbitrary Lagrangian multipliers (since the original problem is itself intractable). One of the contributions of the paper is thus to identify families of Lagrangian multipliers that are (1) amenable to efficient optimization and (2) yield useful bounds for verification. We specifically do so with linear multipliers and exponential multipliers that allow us to improve over SOTA methods for verification of probabilistic specs while remaining tractable.

---

### Official Review · Reviewer_KLAd · 2021-07-23

**Rating:** 7
**Confidence:** 4

**Summary:**

The paper targets the verification of probabilistic specifications of neural networks, e.g., where the specification to a deterministic network can be probabilistic, or for verifying deterministic specifications over a probabilistic neural network. The authors pose the probabilistic verification as an intractable stochastic optimization problem. They leverage functional lagrangian framework to solve the problem approximately. The framework enables the decomposition of the problem into smaller sub-problems which are relatively tractable to solve. The authors experimentally demonstrate the effectiveness of their approach on three specifications against Wicket et al. and obtain more precise results while remaining scalable.

**Limitations And Societal Impact:**

The authors address these concerns adequately in their paper.

**Main Review:**

Originality: The use of functional lagrangian optimization for probabilistic neural networks verification is an original contribution to the best of my knowledge. The authors develop tractable algorithms that exploit the structure of the verification problem for approximately solving the optimization problems. This enables them to verify novel specifications such as distributionally robust OOD detection not possible with prior work.

Quality: The verification framework provided in this work is sound and the experimental results are convincing. In the experiments, the authors use a closed-form solution for the inner optimization problems by instantiating their framework with linear multipliers. This is shown to be equivalent to Dvijotham et al. It is not clear to me whether the closed-form solution is the best solution possible layerwise or an approximation when considering linear multipliers? The Lagrangian dual from Dvijotham et al. does not provide the best layerwise convex relaxation and more precise relaxations were achieved by [1,2].

Clarity: The paper is well motivated and understandable. The authors demonstrate the instantiation of their framework well on an example. I believe that the high-level ideas of the paper should be accessible to a relatively wider audience. However, the authors do not provide details about the CPU cluster used for their experiments in Section 5.1. It is also not clear to me whether the experiments in Section 5.3 were performed on CPUs or GPUs.

Significance: There is relatively less work on verifying probabilistic specifications compared to the deterministic ones. The authors provide a generalized framework for verifying such specifications and show how the existing works can be seen as an instantiation of their framework. I believe that this is a significant contribution and may lead to more specialized verification algorithms with better scalability while maintaining sufficient precision in the future.

More questions:
1. Is the formulation with functional lagrangian presented here the tightest possible or one can define non-decomposable lagrangians which might be more precise?

2. The closed-form solution in C.4 assumes box constraints on x, therefore, processing each neuron independently. If instead, one considered polyhedral dependencies between the input neurons, will the bounds will improve even with linear multipliers?

3. What is the dropout rate for VGG-x networks considered in the experiments?

4. Will the source code of the verifier be publicly available?

References:
1. The Convex Relaxation Barrier, Revisited: Tightened Single-Neuron Relaxations for Neural Network Verification.
2. Beyond the Single Neuron Convex Barrier for Neural Network Certification.

**Time Spent Reviewing:**

7

---

> ### Author Response · Authors · 2021-08-10
> **Thanks for the Comments. Response.**
>
> We thank the reviewer for their detailed comments. We answer point by point below:
>
> ### Main questions
>
> **Q:** Is the formulation with functional lagrangian presented here the tightest possible or one can define non-decomposable lagrangians which might be more precise?
>
> **A:** As shown in Theorem 1, our formulation is exactly tight if any Lagrangian multiplier function can be used. In other words, in the limit where arbitrary multiplier functions can be used, the functional Lagrangian formulation is optimal and cannot be further improved – even through non-decomposable formulations. If, on the other hand, the choice of Lagrangian multiplier functions is restricted (in order to make the problem computationally tractable), then theoretical tightness comparisons are more difficult and remain an open question at this point.
>
> ---
>
> **Q:** The closed-form solution in C.4 assumes box constraints on x, therefore, processing each neuron independently. If instead, one considered polyhedral dependencies between the input neurons, will the bounds will improve even with linear multipliers?
>
> **A:** Considering polyhedral dependencies between the neurons will result in $\mathcal{X}_k$  that are strictly smaller than  $\mathcal{X}_k$ being defined with just box constraints. This however results in harder optimization sub-problems where we cannot leverage a simple closed form solution -- if this harder optimization problem is solved exactly, even with linear multipliers, this should result in a better objective for the sub-problems, and hence the primary verification problem (assuming we are able to find the globally optimal lagrangian parameters).
>
> The interplay between the constraints on $\mathcal{X}_k$, the optimization procedure for upper bounding the sub-problems and the optimization of the lagrangian parameters in practice is an interesting avenue for further understanding and could help improve our method. Further, it is possible that the polyhedral constraints on x are subsumed by the tighter relaxation obtained from more expressive multipliers.
>
> ---
>
> **Q:** What is the dropout rate for VGG-x networks considered in the experiments?
>
> **A:** We used a dropout rate of 0.2 for experiments with the VGG architectures (all details and hyper-parameters are available in Appendix E.1).
>
> ---
>
> **Q:** Will the source code of the verifier be publicly available?
>
> **A:** Yes, we are starting the process of open-sourcing the source code, it will be made available and linked to the paper once published.
>
> ### Questions within comments
>
> **Q:** It is not clear to me whether the closed-form solution is the best solution possible layerwise or an approximation when considering linear multipliers? The Lagrangian dual from Dvijotham et al. does not provide the best layerwise convex relaxation and more precise relaxations were achieved by [1,2].
>
> **A:** The closed-form solution is indeed optimal *given this grouping of layers and relaxation*. Indeed using relu(affine) with linear multipliers has an optimal closed-form solution without any ambiguity.
>
> However one can also consider the use of other layer arrangements, such as affine(relu), in combination with linear multipliers. This requires further relaxation to become tractable, and may be able to express tighter bounds, but also comes at a greater computational cost.
>
> ---
>
> **Q:** However, the authors do not provide details about the CPU cluster used for their experiments in Section 5.1. It is also not clear to me whether the experiments in Section 5.3 were performed on CPUs or GPUs.
>
> **A:** In section 5.3 as well as in section 5.1, the verification of each individual sample was run on the near equivalent of a standard CPU with 4 cores, but the precise specifications of each CPU unit may vary from one sample to another. Experiments from Section 5.2 were run on Tesla P100 GPUs. We will update the submission to make this clearer.

---

### Author Response · Authors · 2021-08-10
**Common Response: Updated Table 2 and Source Code Release**

We would like to thank the reviewers for the feedback.

- For Table 2, we provide estimated upper bounds on the robust accuracy corresponding to an adversarial attack based on sampling weights from the BNN weight-distribution. Below is the updated Table 2:

| Layers  |  Epsilon | Neurons |  BP Acc. (%) | FL Acc. (%) | BP Time (s) | FL Time (s) | Adv Acc (%)  |
| --- | --- | --- | ---| --- | --- | --- | --- |
| 1 | 0.025 | 128  |  43.8  |  65.2  | 1.3 |  +353.3  |  82.6 |
| 1 |  0.025 | 256 | 40.6 | 64.6 | 1.4 | +431.3 | 82.6 |
| 1 | 0.025 | 512 | 35.0 | 57.0 | 1.3 | +357.1 | 82.8 |
| 2 | 0.001 |  256 | 29.4 | 36.9 | 1.6 | +439.6 | 79.4 |
| 2 |  0.001 | 512 | 46.0 | 63.4 | 1.7 | +433.8 | 89.2 |
| 2 | 0.001 | 1024 | 18.4 | 19.6 | 1.6 | +440.9 | 74.8|
------------------------------------------------------------------------------------------

- We will release the source-code for the verifier and the weights for the models publicly along with the manuscript once it is published to ensure reproducibility of our work.

---

### Decision · Program_Chairs · 2021-09-27

**Decision:**

Accept (Spotlight)

**Comment:**

All reviewers unanimously acknowledged the technical contributions in providing mathematical guarantees on neural networks with probabilistic specifications. The authors' responses also addressed the reviewers' concerns. It's a clear accept.